# Adjoint-aided inference of Gaussian process driven differential equations

**Paterne Gahungu**[*]
Quantum Leap Africa
African Institute for Mathematical Sciences, Rwanda
`paterne.gahungu@aims.ac.rw`

**Christopher W. Lanyon**[*]
Department of Computer Science
University of Sheffield, UK
`c.w.lanyon@sheffield.ac.uk`

**Mauricio A. Álvarez**
Department of Computer Science
University of Manchester, UK

**Engineer Bainomugisha**
Department of Computer Science
Makerere University, Uganda

**Michael T. Smith**
Department of Computer Science
University of Sheffield, UK

**Richard D. Wilkinson**
School of Mathematical Sciences
University of Nottingham, UK

## Abstract

Linear systems occur throughout engineering and the sciences, most notably as differential equations. In many cases the forcing function for the system is unknown, and interest lies in using noisy observations of the system to infer the forcing, as well as other unknown parameters. In differential equations, the forcing function is an unknown function of the independent variables (typically time and space), and can be modelled as a Gaussian process (GP). In this paper we show how the adjoint of a linear system can be used to efficiently infer forcing functions modelled as GPs, using a truncated basis expansion of the GP kernel. We show how exact conjugate Bayesian inference for the truncated GP can be achieved, in many cases with substantially lower computation than would be required using MCMC methods. We demonstrate the approach on systems of both ordinary and partial differential equations, and show that the basis expansion approach approximates well the true forcing with a modest number of basis vectors. Finally, we show how to infer point estimates for the non-linear model parameters, such as the kernel length-scales, using Bayesian optimisation.

## 1 Introduction

Linear systems are used as models throughout the sciences and engineering, encompassing a wide range of both ordinary and partial differential equations (including the heat, wave, Schrödinger's, Maxwell's equations etc), as well as systems of linear algebraic equations (such as eigenvalue problems). To fix notation, let $\mathcal{L} : \mathcal{U} \to \mathcal{V}$ be a linear operator between Banach spaces $\mathcal{U}$ and $\mathcal{V}$ (i.e., complete normed spaces, Kreyszig (1991)) . A prototypical linear system is then of the form

$$\mathcal{L}u = f, \tag{1}$$

where $u \in \mathcal{U}$ is the quantity of interest being modelled, and $f \in \mathcal{V}$ is the *forcing function* of the system. Given a fully specified operator $\mathcal{L}$ and forcing function $f$ (and possibly initial and boundary conditions), solving the system for $u$ is referred to as the *forward problem*. Typically, this is a

---

[*]Equal Contribution

36th Conference on Neural Information Processing Systems (NeurIPS 2022).

computationally intensive task. For example, consider modelling air pollution as it moves through the atmosphere. In this case, $\mathcal{L}$ may be a partial differential operator describing the advection, diffusion, and reaction of the pollution, and $f$ will be a function describing the source of the pollution at each location and time. The forward problem refers to computing the concentration, $u$, given the emission sources and will usually require the use of numerical integration methods.

In many applications, both the linear operator $\mathcal{L}$ and the forcing $f$ may not be fully specified, and we may face the statistical task of learning $\mathcal{L}$ and $f$ from noisy observations of $u$:

$$z = h(u) + \epsilon. \tag{2}$$

Here, $z \in \mathbb{R}^n$ are the observations, $h$ the observation operator, $n$ the number of observations, and $\epsilon \in \mathbb{R}^n$ a zero-mean observation error. This is often referred to as the *inverse problem* in applied maths and statistics (Stuart, 2010), or sometimes as a *latent force model* in machine learning (Alvarez et al., 2009). In the air pollution example this would equate to finding the distribution of pollution sources, $f$, given a set of concentration measurements $z$.

We focus on the situation where

1. $f$ is modelled as a Gaussian process (GP) when $\mathcal{V}$ is an infinite dimensional Banach space, or with a Gaussian distribution in the finite dimensional case. E.g., if $\mathcal{L}$ is an ordinary differential operator with independent variable $t$, then $f$ will be an unknown function of $t$.
2. The observation operator $h : \mathcal{U} \to \mathbb{R}^n$ is affine. In the finite dimensional case, this implies $h(u) = Hu + c$ for some constant matrix $H$ and vector $c$, but in the infinite dimensional case (when $\mathcal{U}$ is a space of functions) includes pointwise evaluation of $u$, i.e. $u(x, t)$ for some values of $x$ and $t$, and integral and derivative observations of $u$, e.g. $\int u \, \mathrm{d}x\mathrm{d}t, \frac{\mathrm{d}u}{\mathrm{d}t}$ etc.
3. The observation error $\epsilon$ has a Gaussian distribution. This assumption can be relaxed for maximum likelihood (ML) estimation.

The full specification of the statistical model is then

$$
\begin{aligned}
\mathcal{L}u &= f, & z &= h(u) + \epsilon \\
f &\sim GP(m(\cdot), k(\cdot, \cdot)), & \epsilon &\sim \mathcal{N}_n(0, \sigma^2 I_n)
\end{aligned}
\tag{3}
$$

where $m$ and $k$ are the prior mean and covariance (kernel) functions of the GP. Note that the linear system in Eqs. (3) may include initial and boundary conditions for differential operators. Our aim is to infer $f$ (and possibly $u$, $\mathcal{L}$, and $k$) given $z$ either via

- ML estimation, by solving the constrained optimization problem

$$\min_f \quad (z - h(u))^\top (z - h(u)) + \alpha ||f||_{\mathcal{V}}^2 \qquad \text{s.t.} \ \ \mathcal{L}u = f$$

- Bayesian inference, by computing the posterior distribution

$$\pi(f, u|z) \propto \pi(z|u)\pi(u|f)\pi(f). \tag{4}$$

The prior distribution for $f$, $\pi(f)$, and the regularization parameter, $\alpha$, play a similar role in the two approaches. To solve either problem numerically is likely to require many solves of the forward problem (Eq. 1). For example, for ML we may seek a solution using numerical optimization (e.g. Arellano Jr et al., 2007; Tröltzsch, 2010), whereas with Bayes, we might use an MCMC scheme (e.g. Kopacz et al., 2009; Cotter et al., 2013; Sengupta et al., 2016; Albani et al., 2021) or a variational approach (Chappell et al., 2008). All of these approaches require multiple solves of the forward problem. Here we develop an approach to minimize the computational cost of inference.

## 1.1 Contribution

In this paper we show that implementing an adjoint of the linear system can result in much faster statistical inference. Instead of using numerical approaches to solve either the ML or Bayesian inference problem, we can do inference for $f$ at the cost of $n$ forward model solves, where $n$ is the number of data points. In many (but not all) cases, this will incur a substantially lower computational cost than competing methods, such as MCMC. More specifically, we show that

1. if $f$ depends linearly on parameters $q$, we can estimate $q$ or its distribution analytically, i.e. without resorting to numerical integration methods;
2. if we model $f$ as a Gaussian process, then by using a truncated basis expansion we can efficiently infer the posterior distribution for $f$.

The paper is structured as follows. In the next section we discuss related work before introducing adjoints in Section 3.1. We derive the main results in Section 3.2, and in Section 3.3 we show how linearizing Gaussian processes via a basis expansion reduces inference for GPs in linear systems to simple linear algebra. Finally, in Section 4 we demonstrate the approach on two linear systems: ordinary (ODE) and partial differential equations (PDE).

## 2 Related work

The problem defined by Eqs. (3) is often referred to as a *latent force model* (Alvarez et al., 2009, 2013). Alvarez et al. (2009) showed how the posterior distribution (4) can be computed by using the integral formulation of $\mathcal{L}u = f$, i.e. $u(x) = \int G(x - v)f(v)dv$, where $G(\cdot)$ is the Green's function associated with the differential operator $\mathcal{L}$. Due to the linearity of the integral transform, placing a GP prior over $f$ leads to a joint GP over $f$ and $u$. From this joint GP, the posterior distributions $\pi(u \mid f)$ and $\pi(u)$ can be computed in closed form[2]. However, in many situations, particularly for non-trivial differential equation models, the expressions for the covariances are cumbersome and lead to the use of error functions with complex arguments or functions like the Faddeeva function that can be numerically unstable to compute. Guarnizo & Alvarez (2018) proposed representing $f$ using random Fourier features (RFFs) to reduce the number of integrations necessary to be solved analytically. Here, rather than using Green's functions, we instead use adjoints to write the problem as a linear model and then use a reduced-rank Gaussian process formulation, leading to numerically stable and fast approximations to the posterior distribution.

More specifically, estimation of forcing functions in differential equation models has been extensively studied, for example, in the field of modelling atmospheric advection-diffusion (e.g. Yee, 2008; Borysiewicz et al., 2012; Singh & Rani, 2014; Rajaona, 2016; Yeo et al., 2019) with some authors solving the inverse problem using an adjoint approach in combination with MCMC to compute the posterior distribution (e.g. Yee, 2008; Hwang et al., 2019; Luhar et al., 2020; Albani et al., 2021). Of particular relevance is Hwang et al. (2019) who use a point source model of pollution, and use the adjoint to write the backward finite difference approximation, noting that this can be written as a linear model, where the features are conjugate fields associated with each sensor. MCMC sampling is still a limiting factor, restricting the extension of the approach to more complex situations such as time-varying pollution sources.

Other related work includes the stochastic PDE approach of Lindgren et al. (2011), in which the underlying function $u$ is modelled as a Gauss Markov random field (GMRF), which can then be formulated as a stochastic partial differential equation. This allows finite element methods to be used to efficiently compute the posterior distribution for $u$ given $z$. Similarly, Hartikainen & Särkkä (2010) exploit the link between GMRFs and dynamical systems to convert inference for $u$ to a form in which Kalman filtering methods can be used, which scale linearly with $n$. Sigrist et al. (2015) focus exclusively on the advection diffusion PDE considered in Section 4.2, and use white noise for the forcing function, $f$, to create a stochastic PDE model for $u$. They use a spectral approach, solving the PDE in the Fourier domain, to develop an efficient algorithm for statistical inference. Whilst attractive, it is difficult to generalize this approach from white noise models of $f$ to correlated Gaussian process models. Jidling et al. (2017) show how to infer statistical inference for linearly constrained systems, i.e., where $\mathcal{L}u = 0$. These approaches all model $u$, whereas our focus is on inferring the forcing function $f$.

## 3 Methods

We first recap the definition of adjoints, before deriving our main result illustrating how they can be used to accelerate inference. We then show how using a truncated basis approximation to Gaussian processes allows us to find the GP posterior distribution without resorting to MCMC methods.

### 3.1 Adjoints

Recall that $\mathcal{L} : \mathcal{U} \to \mathcal{V}$ is a linear mapping between Banach spaces $\mathcal{U}$ and $\mathcal{V}$. Let $\mathcal{U}^*$ and $\mathcal{V}^*$ denote the dual spaces of $\mathcal{U}$ and $\mathcal{V}$ (Kreyszig, 1991). We can construct the adjoint to a continuous linear

---

[2]For a more detailed explanation of the Green's function method, please see the supplementary material.

operator $\mathcal{L}$ as follows. Let $v^* \in \mathcal{V}^*$ and define $F : \mathcal{U} \to \mathbb{R}$ by $F : u \mapsto v^*(\mathcal{L}u)$. Then $F$ is a bounded linear functional on $\mathcal{U}$, i.e., $F = u^*$ for some $u^* \in \mathcal{U}^*$. Thus for all $v^* \in \mathcal{V}^*$ we've associated a unique $u^* \in \mathcal{U}^*$:

$$\mathcal{L}^* : v^* \mapsto u^* = v^* \circ \mathcal{L}. \tag{5}$$

We call $\mathcal{L}^*$ the *adjoint* of $\mathcal{L}$, and $\mathcal{L}^*$ is itself a bounded linear operator (Estep, 2004). By construction, we have that for all $u \in U$ and $v^* \in \mathcal{V}^*$

$$(\mathcal{L}^* v^*)(u) = v^*(\mathcal{L}u), \tag{6}$$

a result known as the *bilinear identity*. In the case where $\mathcal{U}$ and $\mathcal{V}$ are also real Hilbert spaces with respect to inner products $\langle \cdot, \cdot \rangle_{\mathcal{U}}$ and $\langle \cdot, \cdot \rangle_{\mathcal{V}}$, then we can identify the dual spaces with their underlying space: by the Riesz representation theorem if $v^* \in \mathcal{V}^*$ then there exists $v \in \mathcal{V}$ such that $v^*(\cdot) = \langle \cdot, v \rangle_{\mathcal{V}}$. In this case, the bilinear identity reduces to its more familiar form

$$\langle \mathcal{L}u, v \rangle_{\mathcal{V}} = v^*(\mathcal{L}u) = (\mathcal{L}^* v^*)(u) = \langle u, \mathcal{L}^* v \rangle_{\mathcal{U}} \tag{7}$$

where we now consider $\mathcal{L}^* : \mathcal{V} \to \mathcal{U}$. We only consider real vector spaces here, resulting in a symmetric inner product.

Generally, the adjoint $\mathcal{L}^*$ will be the same type of operator as $\mathcal{L}$ (e.g. if $\mathcal{L}$ is a differential operator then $\mathcal{L}^*$ will be too), so solving an adjoint system of the form $\mathcal{L}^* v = h$ will have similar computational complexity as solving $\mathcal{L}u = f$. See Estep (2004) for an introduction to adjoints, and Section 4 for examples of adjoint systems.

### 3.2 Benefits of adjoints

How does the development of an adjoint to a linear system help us perform statistical inference for that system? Consider the situation where uncertainty about the unknown forcing function, $f$ in Eq. (1), can be characterized by a linear dependence upon unknown parameters $q$. That is, we can write

$$f(\cdot) = \sum_{m=1}^{M} q_m \phi_m(\cdot). \tag{8}$$

In the infinite dimensional case where $\mathcal{U}$ and $\mathcal{V}$ are spaces of functions on some set $\mathcal{X}$, the $\phi_m$ will also be functions on $\mathcal{X}$. In the finite-dimensional case, the $\phi_m$ will be vectors of fixed length.

In the situation where the observation operator (2) is linear, then we can write the $i^{th}$ observation as $h_i(u) = \langle h_i, u \rangle$ plus noise, for some $h_i \in \mathcal{U}$. Consider the $n$ different adjoint systems

$$\mathcal{L}^* v_i = h_i \text{ for } i = 1, \dots, n.$$

Then using the bilinear identity (7) we have that

$$h_i(u) = \langle h_i, u \rangle = \langle \mathcal{L}^* v_i, u \rangle = \langle v_i, \mathcal{L}u \rangle = \langle v_i, f \rangle,$$

i.e., the $i^{th}$ observation is the inner product between the unknown forcing function $f$ and the solution of the $i^{th}$ adjoint system. At first, the introduction of the adjoint doesn't appear to have helped. To evaluate the likelihood (or sum of squares) we have gone from needing a single solve of the forward problem, to requiring the solution to $n$ adjoint systems. The benefit arises if we now use the assumption of a linear dependence upon the parameters, and linearity of inner products:

$$h_i(u) = \langle v_i, \sum_{m=1}^{M} q_m \phi_m \rangle = \sum_{m=1}^{M} q_m \langle v_i, \phi_m \rangle.$$

The complete observation vector $z$ can then be written as

$$z = \begin{pmatrix} \langle v_1, \phi_1 \rangle & \dots & \langle v_1, \phi_M \rangle \\ \vdots & & \vdots \\ \langle v_n, \phi_1 \rangle & \dots & \langle v_n, \phi_M \rangle \end{pmatrix} \begin{pmatrix} q_1 \\ \vdots \\ q_M \end{pmatrix} + \epsilon = \Phi q + \epsilon \tag{9}$$

where $q \in \mathbb{R}^M$ is the parameter vector, and $\Phi \in \mathbb{R}^{n \times M}$ is the matrix of inner products between the $n$ adjoint solutions and $M$ basis vectors.

This can be recognized as a linear model. Thus, standard results can be used to compute the least squares/ML and Bayesian estimators. For ML, the minimum of $S(q) = (z - h(u))^\top (z - h(u))$ subject to $\mathcal{L}u = f$, is obtained at $\hat{q} = (\Phi^\top \Phi)^{-1} \Phi^\top z$ with $\mathbb{V}\mathrm{ar}(\hat{q}) = \sigma^2 (\Phi^\top \Phi)^{-1}$ in the case where the observation errors $\epsilon_i$ are uncorrelated and homoscedastic with variance $\sigma^2$. Standard results can be used from regularized least squares if we need to regularize $q$.

In a Bayesian setting, if we assume *a priori* that $q \sim \mathcal{N}_M(\mu_0, \Sigma_0)$, then the posterior for $q$ given $z$ (and other parameters) is

$$q \mid z \sim \mathcal{N}_M(\mu_n, \Sigma_n) \tag{10}$$

where

$$\mu_n = \Sigma_n \left( \frac{1}{\sigma^2} \Phi^\top z + \Sigma_0^{-1} \mu_0 \right), \quad \Sigma_n = \left( \frac{1}{\sigma^2} \Phi^\top \Phi + \Sigma_0^{-1} \right)^{-1}. \tag{11}$$

See, e.g., O'Hagan & Forster (2004). Note that the solutions of the adjoint equation, $v_i$, are present in the posterior distribution of $q$ in the matrix $\Phi$.

### 3.3 Inference of Gaussian forcing functions

We now consider the case where the forcing function is given a Gaussian process prior distribution:

$$f(\cdot) \sim GP(m(\cdot), k(\cdot, \cdot)), \tag{12}$$

where $m(\cdot)$ and $k(\cdot, \cdot)$ are the prior mean and covariance functions respectively (Rasmussen & Williams, 2006). Our approach is to use a reduced-rank representation of the GP, as in Eq. (8), derived by truncating an expansion for $k$ of the form

$$k(x, x') = \sum_{m=1}^{\infty} \phi_m(x)\phi_m(x'). \tag{13}$$

There are many possible choices for the basis vectors $\phi_i(\cdot)$, including the Karhunen-Loève (KL) (Deheuvels & Martynov, 2008), Laplacian (Solin & Särkkä, 2020; Coveney et al., 2020; Borovitskiy et al., 2020), and random Fourier feature (RFF) (Rahimi et al., 2007) expansions. The Karhunen Loève basis is formed by finding the spectral expansion of the integral operator defined in Mercer's theorem:

$$T_k f(x) = \int_{\mathcal{X}} k(x, x') f(x') \mathrm{d}x'.$$

In the case where $\dim(x)$ is small (such as the ODE example below), the eigenfunctions of $T_k$ are easy to compute either analytically (Rasmussen & Williams, 2006, Section 4.3.2, p116) or numerically (Greengard & O'Neil, 2021). Using the eigenfunctions of $T_k$ gives the $L^2$-optimal approximation (Kosambi, 2016), but the eigenfunctions can be difficult to compute even in three dimensions. A simpler approach that extends easily to higher dimensional problems, is to use RFFs (Rahimi et al., 2007). This relies upon Bochner's theorem, which can be used to express stationary kernels $k(x, x') := k(x - x')$ as the Fourier transform of a positive measure $p$

$$k(x - x') = \int \exp(-iw(x - x')) \mathrm{d}p(w).$$

If we use an isotropic exponentiated quadratic (EQ) kernel $k$, then the measure $p$ corresponds to a multivariate Gaussian, and Rahimi et al. (2007) give expressions for the Fourier feature basis vectors that can be used to approximate the Gaussian process:

$$k(x - x') = \tau^2 \exp(-\frac{1}{2\lambda^2}(x - x')^\top (x - x')) \quad \Rightarrow \quad \phi_m(x) = \sqrt{\frac{2\tau^2}{M}} \cos(\frac{1}{\lambda} w_m^\top x + b_m) \tag{14}$$

where $w_m \sim N(0, I)$ and $b_m \sim U(0, 2\pi)$. Substituting this $\phi_m$ into equation 8 allows us to write the forcing function $f$ as a truncated Gaussian process with an EQ kernel (as such the true Gaussian process is never calculated). The extension to anisotropic kernels is straight-forward.

Although the RFF expansion will require more basis vectors (a larger $M$) than the KL expansion to achieve the same accuracy, the computational complexity of our adjoint approach is dominated by the number of adjoint solves, not the number of features (which only affects the cost of computing the relatively low-cost Eq. 13), and so including more terms has a minor effect on overall computational cost.

### 3.4 Time complexity of Algorithm 1

Let $G$ be the number of grid elements, $n$ the number of observations, and $M$ the number of features. There are five stages in our algorithm:

1. Solving the $n$ adjoint systems, which requires $\mathcal{O}(Gn)$ operations.
2. Computing each basis vector, $\phi$, over the grid for each feature requires $\mathcal{O}(GM)$ operations.
3. Computing the matrix $\Phi$, requires $\mathcal{O}(GMn)$ operations.
4. Finding $\Phi^\top \Phi$ requires $\mathcal{O}(nM^2)$ operations.
5. Finally solving the matrix inverse will require $\mathcal{O}(M^3)$ operations.

This results in an algorithm that scales linearly in the number of data points, $n$. Empirically, for the problems we've experimented with, we found computing $\phi$ over the large grid was the most time consuming step. The overall time complexity is $\mathcal{O}(GMn) + \mathcal{O}(nM^2) + \mathcal{O}(M^3)$, but note that the constants associated with each term are important in most problems.

## 4 Experiments

### 4.1 Ordinary differential equation (ODE)

Consider the non-homogeneous linear ODE:

$$p_2 \frac{\mathrm{d}^2 u}{\mathrm{d}t^2} + p_1 \frac{\mathrm{d}u}{\mathrm{d}t} + p_0 u = f(t) \tag{15}$$

on the domain $[0, T]$ with initial conditions $u(0) = u'(0) = 0$. The right hand side, $f(t)$, is the unknown forcing function that we wish to estimate, and $p_0, p_1, p_2$ are parameters in the linear operator. We model $f$ as a zero-mean GP with an EQ kernel (Eq. 14).We assume observations are obtained as noisy averages over short time windows:

$$h_i(u) = \int_{t_i}^{t_i+\Delta t} \frac{1}{\Delta t} u(t)\mathrm{d}t = \langle u, \tilde{h}_i \rangle \tag{16}$$

where $\tilde{h}_i$ is the indicator function $\tilde{h}_i(t) = \mathbb{I}_{[t_i, t_i+\Delta t]}(t)$. This includes direct measurements of $u(t_i)$ if we set $\tilde{h}_i(t) = \delta(t - t_i)$, the Dirac delta function. To generate synthetic data, we simulate a realization $f$ from the GP model, solve Eq. (15) for $u(t)$, and then simulate $n$ observations from Eq. (16) with $T = 1, p_2 = 0.5, p_1 = 1, p_0 = 5, \Delta t = \frac{T}{n}, t_i = \frac{iT}{n}$, adding zero-mean Gaussian noise with standard deviation $\sigma = 0.1$. For simplicity, we solve the ODE with a simple forward Euler approximation, but higher order schemes can and should be used in real applications. We approximate the GP using Eq. (8), using 200 RFFs generated using Eq. (14) with $\lambda = \sqrt{0.6}$ and $\tau^2 = 4$. The linear operator in this case is

$$\mathcal{L}u = \left( p_2 \frac{\mathrm{d}^2}{\mathrm{d}t^2} + p_1 \frac{\mathrm{d}}{\mathrm{d}t} + p_0 \right) u.$$

To derive the adjoint operator we use the bilinear identity (Eq. 7), and integrate by parts twice:

$$\langle \mathcal{L}u, v \rangle = \int_0^T (\mathcal{L}u)v\mathrm{d}t = \int_0^T (p_2\ddot{u} + p_1\dot{u} + p_0 u)v\mathrm{d}t = \int_0^T (p_2\ddot{v} - p_1\dot{v} + v)u\mathrm{d}t = \langle u, \mathcal{L}^* v \rangle$$

when $v(T) = \dot{v}(T) = 0$. So the adjoint of $\mathcal{L}$ is

$$\mathcal{L}^* v = \left( p_2 \frac{\mathrm{d}^2}{\mathrm{d}t^2} - p_1 \frac{\mathrm{d}}{\mathrm{d}t} + p_0 \right) v.$$

Note that rather than an initial condition, the adjoint has a final condition: to solve the system we have to integrate backwards in time from $t = T$ to $t = 0$. See Algorithm 1.

Fig. 1 shows the effects of the number of training points $n$ and the number of features $M$ on the posterior distribution of the forcing function. As expected, more data results in a more confident

**Algorithm 1** Computing the posterior distribution of $q$

---
**for** $i = 1 \ldots n$ **do**
    Solve adjoint system $\mathcal{L}^* v_i = \tilde{h}_i$ with appropriate final and boundary conditions.
**end for**
**for** $m = 1 \ldots M$ **do**
    Sample an RFF basis vector $\phi_m$ using Eq. (14).
    **for** $i = 1 \ldots n$ **do**
        Compute $[\Phi]_{im} = \langle v_i, \phi_m \rangle$.
    **end for**
**end for**
Compute the posterior distribution for $q$ using Eqs. (10) and (11).

---

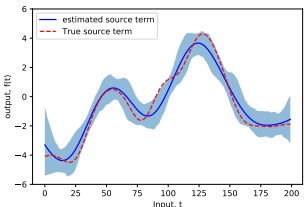 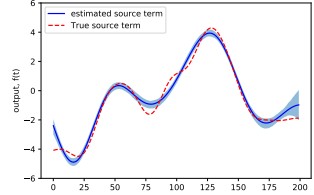 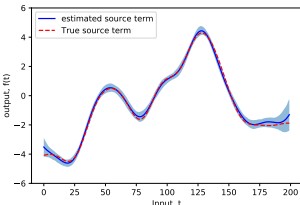

Figure 1: Posterior distribution for the unknown forcing function (with 95% credibility interval). True forcing in red. The number of training points and features are, (left) $n = 10$ and $M = 100$; (middle) $n = 100$ and $M = 10$; (right) $n = 100$ and $M = 100$. The overconfident and wrong posterior when $M = 10$ is a consequence of the model being heavily misspecified.

posterior. Note though the danger of using too few features: with $M = 10$ the approximation of $f$ has limited expressive power and cannot capture the true form of $f$, i.e., the model is heavily misspecified. This can result in the uncertainty collapsing upon the most likely, but wrong, value. This can be difficult to spot, so users should check the sensitivity of the posterior with respect to $M$ (which can be done with no additional forward solves).

In the supplementary material we provide a comparison between the adjoint method, the Green's function methods as in Alvarez et al. (2009) and Guarnizo & Alvarez (2018), and a vanilla Gaussian process when applied to this ODE problem.

## 4.2 Partial differential equation (PDE)

We now demonstrate the approach on a PDE in which there are two spatial variables, $x \in \mathcal{X} \subset \mathbb{R}^2$, and time, $t \in [0, T] \subset \mathbb{R}$, so that the solution $u \equiv u(x, t)$ is a function of three independent variables. We consider the advection-diffusion equation

$$\frac{\partial u}{\partial t} + p_1 \cdot \nabla u - \nabla \cdot (p_2 \nabla u) = f \text{ in } \mathcal{X} \times [0, T] \tag{17}$$

with initial and boundary conditions $u(x, 0) = 0$ for $x \in \mathcal{X}$ and $\nabla_n u = 0$ for $x \in \partial \mathcal{X}$. Here, the unknown forcing function $f \equiv f(x, t)$ is a function of space and time, and we model it as a zero-mean Gaussian process, $f(x, t) \sim GP(0, k((x, t), (x', t'))$ with EQ kernel $k$ (Eq. 14). We use an RFF approximation to $k$ with random weights $w_m \sim \mathcal{N}_3(0, I)$ and $b_m \sim U(0, 2\pi)$.

Observations are assumed to arise from *sensors* which take an average of $u(x, t)$ over a small spatial and temporal window $\mathcal{R}_i \times \mathcal{T}_i \subset \mathcal{X} \times [0, T]$:

$$z_i = \langle u, \tilde{h}_i \rangle + e_i \quad \text{with} \quad \tilde{h}_i = \begin{cases} \frac{1}{|\mathcal{R}_i| . |\mathcal{T}_i|} & \text{if } x \in \mathcal{R}_i \text{ and } t \in \mathcal{T}_i \\ 0 & \text{otherwise.} \end{cases}$$

The adjoint of the linear operator $\mathcal{L}u = \frac{\partial u}{\partial t} + p_1 . \nabla u - \nabla \cdot (p_2 \nabla u)$ is

$$\mathcal{L}^* v = -\frac{\partial v}{\partial t} - p_1 . \nabla v - \nabla \cdot (p_2 \nabla v).$$

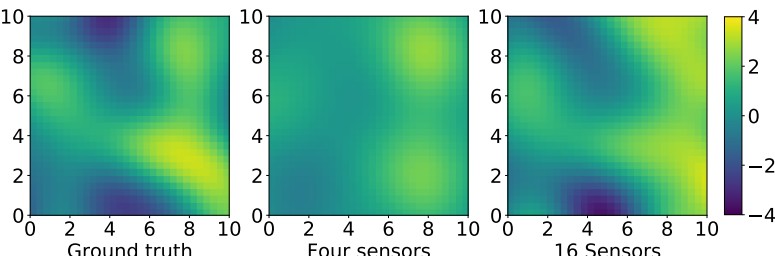

Figure 2: Spatial maps of the forcing function at a single slice: $f(x, 5)$. Left shows ground truth, middle shows the posterior mean with 4 sensors, right shows the posterior mean using 16 sensors.

Table 1: Posterior mean and standard deviation of the $q$ parameters estimated by MCMC and the adjoint method.

|  | $q_1$ | $q_2$ | $q_3$ | $q_4$ | $q_5$ |
|---|---|---|---|---|---|
| MCMC | -3.62 (0.02) | -0.64 (0.01) | 1.68 (0.01) | -0.09 (0.01) | 4.31 (0.02) |
| Adjoint | -3.62 (0.02) | -0.64 (0.01) | 1.68 (0.01) | -0.09 (0.01) | 4.31 (0.02) |

Our adjoint-aided approach then requires the solution of

$$\mathcal{L}^* v_i = \tilde{h}_i \text{ in } \mathcal{X} \times [0, T] \text{ for } i = 1, \ldots, n.$$

The final and boundary conditions on the adjoint system are

$$v_i(x, T) = 0 \text{ for } x \in \mathcal{X}, \quad \text{and} \quad p_1 v_i(x, t) + p_2 \nabla v(x, t) = 0 \text{ for } x \in \partial\Omega \text{ and } t \in [0, T].$$

For details of the adjoint derivation see the supplementary material and Estep (2004).

Inference then proceeds as before. After solving the $n$ adjoint systems, we compute the inner product of these solutions with the RFF basis vectors to form the matrix $\Phi$ (Eq. 9). We can then use Eq. (11) to compute the posterior with minimal additional computational cost.

Data was simulated on the spatial domain $\mathcal{X} = [0, 10]^2$ for $t \in [0, 10]$ by first randomly generating a forcing function $f(x, t)$ (generated from a GP using an EQ kernel with $\lambda = 2$, $\tau^2 = 2$), and then solving the forward problem (Eq. 17) to find $u(x, t)$ using PDE parameters $p_1 = (0.4, 0.4)$ and $p_2 = 0.01$. We generate $n$ observations using sensors that record averages over short time windows equally spaced across the domain $[0, 10]$ at the locations shown in Fig. 3. Zero-mean Gaussian distributed noise is added to the true sensor readings with standard deviation $\sigma = 0.05$ (note that this is relatively small compared to the signal, which can often create problems for sampling methods). We then use Algorithm 1 to calculate the posterior distribution for $q$, hence giving the posterior for $f$. By sampling forcing functions from this posterior and simulating forward, we can evaluate the posterior predictive accuracy of the model.

To validate the posterior estimates from the adjoint method, we also used MCMC to compute the posterior distribution for the PDE model using just $M = 10$ RFFs. Table 1 shows the posterior mean and variance of the first 5 $q$ parameters determined using both methods, which can be seen to be in close agreement. Fig. 7 in the supplementary material shows the trace plots. We used a batch-update random-walk Metropolis-Hastings (MH) sampler, which failed to converge (after an hour of computation) when using a larger number of features. Even with $M = 10$, we still required 10,000s of forward model evaluations to reach convergence (whereas in comparison, the adjoint method required just 75 forward model solves in this case).

To illustrate the effects of sensor density, observations were generated for five time windows using arrays of 4 and 16 sensors at the locations shown in Fig. 3, i.e., a total of either $n = 20$ or $n = 80$ observations. The ground truth forcing and inference results (with $M = 200$) are shown in Fig. 2. As expected, more sensors results in improved estimates. Fig. 3 shows the posterior standard deviation for $f$ in the 4 and 16 sensor cases. Here advection occurs parallel to the $y = x$ line as $p_1 = (0.4, 0.4)$ (i.e., as if there is wind blowing from the south west). We can see that standard deviation is smallest *upwind* of the sensors, with more uncertainty *downwind*, as expected.

To investigate the effects of varying feature and sensor numbers we performed a posterior predictive check using held-out data and used Monte Carlo estimation to calculate the posterior predictive mean

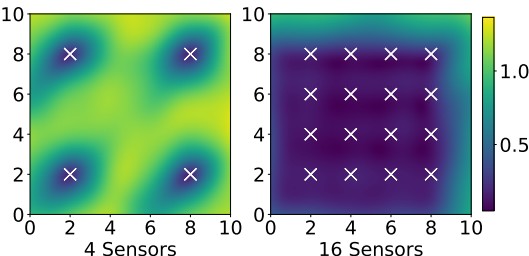

Figure 3: Posterior standard deviation for the examples shown in Fig. 2. The left image is for the case where 4 sensors are used, and the right for 16 sensors. Sensor locations are shown as white crosses.

Table 2: The median MSE as a function of number of sensors and RFFs. The ground truth was generated using a GP with $\lambda = 2$, $\tau^2 = 2$ and PDE parameters $p_1 = (0.01, 0.01)$, $p_2 = 0.01$. The MH algorithm did not converge after 20,000 iterations for 50 or more RFFs. The numbers brackets are the 95% confidence intervals computed from 10 repeated experiments (for the adjoint approach) and 5 (for MH).

| Sensors | Features | | | | | |
|---|---|---|---|---|---|---|
| | 10 | 50 | 100 | 200 | 300 | 500 |
| 1 | 3.42 (2.82,4.39) | 3.27 (3.13,3.38) | 3.24 (3.10,3.37) | 3.27 (3.17,3.44) | 3.24 (3.12,3.36) | 3.27 (3.17,3.35) |
| 4 | 7.12 (1.57,28.81) | 2.39 (2.06,2.62) | 2.41 (2.13,2.60) | 2.45 (2.32,2.57) | 2.50 (2.41,2.69) | 2.53 (2.32,2.60) |
| 9 | 2.38 (1.41,4.40) | 2.12 (1.48,3.98) | 1.70 (1.49,2.07) | 1.48 (1.40,1.72) | 1.47 (1.32,1.61) | 1.45 (1.40,1.50) |
| 16 | 1.73 (1.23,3.28) | 3.99 (2.32,10.90) | 2.18 (1.72,3.54) | 1.3 (1.02,1.68) | 1.12 (0.98,1.37) | 1.12 (1.02,1.21) |
| 25 | 1.35 (1.19,3.09) | 8.93 (4.92,39.86) | 4.36 (2.53,8.20) | 1.86 (1.43,2.75) | 1.35 (1.07,1.81) | 1.05 (0.89,1.45) |
| 25 (MH) | 3.27 (1.73,6.12) | - | - | - | - | - |

squared error (MSE). Table 2 gives the MSE as the number of sensors and RFFs vary. As both increase, so does the accuracy of our estimates. In general, the accuracy depends upon a variety of factors, including the PDE parameters (ratio of diffusion to advection), kernel parameters (decreasing lengthscale makes the problem more challenging), and sensor locations. The speed and efficiency of the proposed adjoint-aided approach allows us to investigate these effects in a way that would not be possible if we were using MCMC (as each estimate of the posterior requires tens of thousands of simulator evaluations, rather than just the $n$ evaluations required for the adjoint-aided approach).

Although not our focus here, we note that we can infer the remaining parameters $p$ (PDE) and $(\tau^2, \lambda)$ (GP hyperparameters) in a variety of ways. For example, using Bayesian optimization we can find the maximum likelihood estimates of these parameters with relatively minimal computational cost. See the supplementary material for details and examples.

In the supplementary material we include some examples of the effect of varying the Gaussian process lengthscale on inference quality, an example in which we apply our method to an advection diffusion problem using the Roundhill II dataset (Cramer & Record, 1957) and apply our method to shift operators (a non-differential linear operator).

### 4.3   Real time cost analysis

We compared the run time of the adjoint method to a basic MH MCMC algorithm (recorded on a laptop with 16GB RAM and an Intel i7-1065G7 CPU @ 1.50 GHz). Exact inference of the posterior of $f$ in the ODE model using the adjoint method on a 100 element temporal grid with 100 observations using 100 RFFs took $569 \pm 72.5$ ms. This is approximately the time it took to perform a single iteration of the MH algorithm. Inference of the posterior of $f$ in the PDE model using the adjoint method on a $50 \times 30 \times 30$ spatiotemporal grid with 100 observations using 100 RFFs took $3.32 \pm 0.18$ s. In this time we were able to run $11 \pm 2$ iterations of the MH algorithm.

The Metropolis Hastings algorithm is comparatively inefficient due to the matrix multiplication required to compute the forcing function $f$ from a given $q$ using the basis vectors (see equation 8). In the adjoint method this only needs to be calculated once, but is required at each step of the MH algorithm.

Whilst more sophisticated MCMC implementations will achieve faster convergence than random walk samplers, any MCMC algorithm will require 10,000s of forward model evaluations to approximate the same posterior the adjoint method can compute exactly[3] in a fraction of the time.

## 5  Discussion

This work was motivated by the problem of inferring the distribution of spatially and temporally varying sources of pollution across a city from noisy observations using a model of the pollution's atmospheric transport. Estimating the pollution concentration and its sources can help local authorities reduce the population's exposure, and to motivate policy interventions. Linear systems such as this, are typical of the type of challenge faced throughout the sciences and engineering, and still pose computational challenges that make principled statistical inference almost intractable, or if they are tractable, the computational cost (e.g. of using MCMC) is such that only a limited range of models and situations can be analysed. In this paper we developed an approach that results in conjugate Bayesian inference of an unknown GP forcing function, with a computational cost that scales linearly with the number of observations, $n$. As $n$ increases, the approach may eventually require more computation than competitor methods such as MCMC (which in theory has cost independent of $n$), but given that MCMC typically requires $10^5 - 10^6$ iterations even for low-dimensional problems[4], there is still a large range of problems for which an adjoint may be beneficial. In our PDE example, the adjoint-aided approach required orders of magnitude fewer PDE solves than MCMC. We should note that a key limitation of our approach is that it only applies to linear systems and cannot be used to determine forcing functions for non-linear systems such as the Navier-Stokes equations.

We only briefly touched upon the problem of estimating the operator's non-linear parameters $p$ (e.g., advection and diffusion rates) and the GP hyperparameters. *Adjoint sensitivity* methods (see, e.g. Bradley, A. M., 2010; Margossian, 2019; Yashchuk, 2020) can be used to estimate gradients of the log-likelihood with respect to the non-linear parameters in a numerically stable alternative to automatic-differentiation frameworks (such as TensorFlow), which can be unstable when back propagating through long iterative loops. Access to gradient information allows inference of the additional parameters to be performed efficiently within the preferred statistical paradigm (e.g. maximum likelihood using Bayesian optimization - see the supplementary information; Hamiltonian Monte Carlo (Neal, 2011); or a variational autoencoder (Kingma & Welling, 2014) framework etc.).

Finally, there are many ways in which this approach may be accelerated, for example, by the use of intelligent numerical solvers that reuse solution trajectories and adaptive step size solvers, multi-fidelity methods that use varying grid sizes, and stochastic approaches which use only a subset of the data at each stage. The core computational tasks in our approach (solving the adjoint systems) are embarrassingly parallelisable, enabling easy deployment on HPC facilities if required.

**Software:** The algorithm has been implemented (for both the ODE and PDE problems) in a Python module available at https://github.com/SheffieldML/advectionGP. The repository also contains Jupyter notebooks that are used to produce the figures and tables in this paper.

## Acknowledgements

This work was directly funded by EPSRC projects EP/T00343X/1 and EP/T00343X/2. In addition, EB was supported by funding from Google.org and RW by EPSRC projects EP/P010741/1, EP/W000091/1, and EP/X012603/1.

---

[3]Here "exactly" refers to our ability to write down and compute the posterior for a given basis truncation of the GP, rather than approximating the posterior with MCMC.

[4]In the best case scenario of independent Gaussian posteriors, the number of iterations required for random walk Metropolis Hasting scales as $\mathcal{O}(M^2)$ (Gelman et al., 1997) and $\mathcal{O}(M^{\frac{5}{4}})$ for Hamiltonian Monte Carlo, (Beskos et al., 2013) where $M$ is the dimension of the parameter, with each iteration requiring a solve of the forward problem. For more complex problems, scaling rates will be worse.

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
