**Supplementary material for** *Adjoint-aided inference of Gaussian process driven differential equations*

## Latent force models using Green's functions

Existing approaches to latent force models rely upon the concept of a Green's function (e.g. Higdon, 2002; Boyle & Frean, 2005; Alvarez et al., 2009, 2013; Guarnizo & Alvarez, 2018). Here, we briefly describe this approach and how it links to our adjoint-aided approach, and discuss the advantages and disadvantages of both methods. Consider the linear system

$$\mathcal{L}u = f \qquad \text{for } x \in \Omega \tag{18}$$
$$u = 0 \qquad \text{for } x \in \partial\Omega.$$

Here, $\mathcal{L}$ is assumed to be a differential operator, and the solution $u$ is a function of $x$ with domain $\Omega$. The Green's function for this system, $G_y(x)$, satisfies

$$\mathcal{L}^* G_y(x) = \delta_y(x) \qquad \text{for } x \in \Omega \tag{19}$$
$$G_y(x) = 0 \qquad \text{for } x \in \partial\Omega$$

where $\delta_y(x) = \delta(x - y)$ is the Dirac delta function, and $\mathcal{L}^*$ is the adjoint of $\mathcal{L}$. Once we have determined a Green's function, solution of the original problem (18) can be found by computing the convolution of $G$ with $f$:

$$
\begin{aligned}
u(y) &= \langle \delta_y, u \rangle && \text{by definition of Dirac delta} \\
&= \langle \mathcal{L}^* G_y, \ u \rangle && \text{by Eq. (19)} \\
&= \langle G_y, \ \mathcal{L}u \rangle && \text{by definition of the adjoint} \\
&= \langle G_y, \ f \rangle && \text{by Eq. (18)} \\
&= \int G_y(x) f(x) \mathrm{d}x.
\end{aligned}
$$

The standard approach to latent force models then assumes $f$ is a Gaussian process, $f \sim GP(0, k)$, and uses the linearity of this expression and the closure of Gaussian processes under linear operations (Rasmussen & Williams, 2006) to conclude that $u$ is also distributed as a Gaussian process,

$$u \sim GP(0, k_u)$$

with covariance function

$$k_u(y, y') = \int G_y(x) \int G_{y'}(x') k(x, x') \mathrm{d}x' \mathrm{d}x. \tag{20}$$

For some forms of the kernel $k$, e.g. the exponentiated quadratic kernel, it is possible to evaluate these integrals analytically when $G$ is known. Alternatively, we can resort to numerical integration to evaluate Eq. (20), for example, using random Fourier features (Guarnizo & Alvarez, 2018). Other works have represented $G$ using a polynomial series (Guarnizo & Álvarez, 2018) or have put another GP prior over $G$ (Tobar et al., 2015).

When the Green's function is known for a given system, this approach can work efficiently and may perform as well or better than the adjoint-aided approach. See Cole (2000) for a comprehensive list of Green's functions. However, for many systems (particularly operators with spatially/temporally varying coefficients) the Green's functions are not analytically computable. For diagonalizable operators, we can try to find the eigenfunctions of $\mathcal{L}$, i.e., $\lambda_i$, and $\phi_i(x)$ such that $\mathcal{L}\phi_i = \lambda_i \phi_i$, then we can write

$$G_y(x) = \sum_{i=1}^{\infty} \frac{1}{\lambda_i} \phi_i(x) \phi_i(x').$$

If we can estimate $\lambda_i$ and $\phi_i(x)$ numerically (i.e., by numerically solving the differential equation $\mathcal{L}\phi_i = \lambda_i \phi_i$ on some computational mesh) we can then truncate this sum and form a numerical approximation of $G_y(x)$. But in this case, we would then need to use our numerical approximation of $G$ in a further numerical approximation of the integral in Eq. (20) which can easily lead to numerical

instabilities and low accuracy. In addition, not all differential operators are diagonalizable (i.e., admit a basis of eigenfunctions), for example, operators which are not self-adjoint.

In contrast, our adjoint-aided approach relies solely on the existence of the adjoint operator $\mathcal{L}^*$ and our ability to solve adjoint systems numerically. To do this, we can deploy modern finite element solvers that are efficient, stable, and offer good error-control. A full numerical analysis of the respective errors of the two approaches is beyond the scope of this paper, and would necessarily be specialized to the implementation details of all the particular numerical algorithms used.

In summary, we would recommend that in the special case where $G$ is known and Eq. (20) is tractable, that a Green's function approach be used. In other situations, the ease of the adjoint approach introduced here is likely to be an attractive alternative both in terms of accuracy, numerical stability, and ease of implementation.

## Comparison to competing methods

We conducted a comparison between the adjoint method, the Green's function method and a classical Gaussian process on the ordinary differential equation model presented in section 4.1. Observations were taken at 20 time points over $t \in [0, 10]$ with a grid resolution of 200.

- The Gaussian process had a mean squared error of 0.0055 between the true output and the inferred output.
- The Green's function method (as in Alvarez et al. (2009)) had an MSE of 0.0051 for the output error and 0.0860 for the source error.
- The Green's function method with random Fourier features (as in Guarnizo & Alvarez (2018)) achieved the following MSEs:
  - 20 features: Source MSE of 0.099 and output MSE of 0.0058
  - 200 features: Source MSE of 0.0927 and output MSE of 0.0055.
  - 500 features: Source MSE of 0.0856 and output MSE 0f 0.0052.
  - 2000 features: Source MSE of 0.0861 and outpute MSE of 0.0051.
- The adjoint method with M=2000 random Fourier features had an MSE of 0.0056 between the ground truth concentration and the inferred concentration and an MSE of 0.079 between the ground truth and inferred sources.

All three methods achieve a similar quality of inference over the system output. This is to be expected as all three methods utilise a similar statistical model. For larger numbers of features ($M \sim 200$) the adjoint method and the GP predicted the system response with similar accuracy. It should be noted that by using a classical GP approach it is not possible to infer the unknown forcing function, $f$, which is one of the key advantages of the adjoint method. The Green's function method also performs to a similar level of accuracy as the adjoint method, though the Green's function method with Fourier features appears to perform better at low numbers of features for this particular test case.

## Derivation of the Advection Diffusion Adjoint Equation

Consider the advection diffusion operator discussed in Section 4.2:

$$\mathcal{L}u = \frac{\partial u}{\partial t} + \mathbf{p}_1 \cdot \nabla u - \nabla \cdot (p_2 \nabla u) \text{ in } \mathcal{X} \times [0, T] \tag{21}$$

with initial condition

$$u(x, 0) = 0 \text{ for all } x \in \mathcal{X} \tag{22}$$

and Neumann boundary condition

$$\nabla_n u = 0 \text{ for x} \in \partial\mathcal{X}, \tag{23}$$

where $\partial\mathcal{X}$ is the boundary of $\mathcal{X}$, $\nabla_n u = \nabla u \cdot \hat{\mathbf{n}}$ denotes the normal derivative of $u$, with $\hat{\mathbf{n}}(x)$ the outward facing normal of $\partial\mathcal{X}$ at $x$. Let $\Omega = \mathcal{X} \times [0, T]$ denote the spatial temporal domain of $u$.

The adjoint of the system defined by Eqs (21–23) will depend on both the differential operator, and the specific initial and boundary conditions imposed. To derive this, we need to find a linear operator $\mathcal{L}^*$ and a set of boundary conditions so that the bilinear identity

$$\langle \mathcal{L}u, v \rangle = \langle u, \mathcal{L}^*v \rangle$$

is satisfied for all sufficiently smooth functions $u$ and $v$ with compact support in $\Omega$. Let $v$ be such a function, and consider

$$\langle \mathcal{L}u, v \rangle = \int_\Omega \left( \frac{\partial u}{\partial t} + \mathbf{p}_1 \cdot \nabla u - \nabla \cdot (p_2 \nabla u) \right) v \, d\Omega. \tag{24}$$

In the derivation below, we'll assume $\mathbf{p}_1$ and $p_2$ are constant, and follow the general steps outlined in Estep (2004). As in the ODE example, the derivation essentially relies upon repeated application of integration by parts. For the first term in Eq. (24):

$$\int_\Omega \frac{\partial u}{\partial t} v \, d\Omega = \int_\Omega \frac{\partial}{\partial t}(uv) - u \frac{\partial v}{\partial t} \, d\Omega$$

$$= \int_\mathcal{X} \int_0^T \frac{\partial}{\partial t}(uv) \, dt \, dx - \int_\Omega u \frac{\partial v}{\partial t} \, d\Omega$$

$$= \int_\mathcal{X} u(x,T)v(x,T) - u(x,0)v(x,0) \, dx - \int_\Omega u \frac{\partial v}{\partial t} \, d\Omega.$$

For the second term in Eq. (24):

$$\mathbf{p}_1 \cdot \int_\Omega v \nabla u = \mathbf{p}_1 \cdot \left( \int_\Omega \nabla(uv) \, d\Omega - \int_\Omega u \nabla v \, d\Omega \right)$$

$$= \mathbf{p}_1 \cdot \left( \int_0^T \int_\mathcal{X} \nabla(uv) \, dx \, dt - \int_\Omega u \nabla v \, d\Omega \right)$$

$$= \mathbf{p}_1 \cdot \left( \int_0^T \oint_{\partial \mathcal{X}} uv \hat{\mathbf{n}} \, dx \, dt - \int_\Omega u \nabla v \, d\Omega \right)$$

where the first equality uses the vector product rule, and the third the divergence theorem. For the third term in Eq. (24) we have

$$p_2 \int_\Omega v \nabla \cdot \nabla u \, d\Omega = p_2 \left( \int_0^T \oint_{\partial \mathcal{X}} v \nabla u \cdot \hat{\mathbf{n}} \, dx \, dt - \int_\Omega \nabla v \cdot \nabla u \, d\Omega \right).$$

We can then repeat this process on the final term above

$$\int_\Omega \nabla v \cdot \nabla u \, d\Omega = \int_0^T \oint_{\partial \mathcal{X}} u \nabla v \cdot \hat{\mathbf{n}} \, dx \, dt - \int_\Omega u \nabla \cdot \nabla v \, d\Omega.$$

Combining all of these terms together gives

$$\langle \mathcal{L}u, v \rangle = \int_\Omega \left( -\frac{\partial v}{\partial t} - \mathbf{p}_1 \cdot \nabla v - \nabla \cdot (p_2 \nabla v) \right) u \, d\Omega$$

$$+ \int_\mathcal{X} u(x,T)v(x,T) - u(x,0)v(x,0) \, dx$$

$$+ \int_0^T \oint_{\partial \mathcal{X}} uv \mathbf{p}_1.\hat{\mathbf{n}} - p_2 v \nabla u \cdot \hat{\mathbf{n}} + p_2 u \nabla v \cdot \hat{\mathbf{n}} \, dx \, dt$$

$$= \langle u, \mathcal{L}^*v \rangle + \text{boundary terms}.$$

As in the ODE case, we then choose the boundary and initial conditions on $v$ to make the boundary terms above vanish. Firstly, as $u(x,0) = 0$ for all $x$, setting the final condition $v(x,T) = 0$ for all $x$ eliminates the first boundary term. Secondly, as $\nabla u \cdot \hat{\mathbf{n}}$ is 0 on the boundary (from the boundary conditions on $u$, Eq. 22), the third term also vanishes. Finally, to set the remainder of the boundary integral to 0 we assume $\mathbf{p}_1 v + p_2 \nabla v = 0$ for all $x \in \partial \mathcal{X}$.

Thus our adjoint operator is

$$\mathcal{L}^* v = -\frac{\partial v}{\partial t} - p_1 \nabla \cdot v - \nabla \cdot (p_2 \nabla v) \tag{25}$$

with final condition

$$v(x, T) = 0 \text{ for all } x \in \mathcal{X}$$

and mixed condition

$$\mathbf{p}_1 v + p_2 \nabla v = 0 \text{ for } x \in \partial \mathcal{X}.$$

Note that when solving the original and adjoint systems numerically, checking that the bilinear identity does indeed hold is a useful validation of the derivation and PDE solvers.

## PDE Inference Examples

Various factors effect the quality of source inference when using the adjoint method. These include the number of random Fourier features (RFFs), $M$, the number of observations, $n$, the locations of the sensors, and the ratio of the ground truth source lengthscale, $\lambda$, to the size of the domain. In Section 4.2 we investigated the effect of changing the values of $n$ and $M$ (see Table 2) for a system with a fixed lengthscale. Here we briefly illustrate the effect of changing $\lambda$. We consider a $10 \times 10 \times 10$ grid in space and time and two scenarios:

1. 100 sensors arranged in a grid, with readings at 10 points in time, using 1000 RFFs to infer the source ($n = 1000$, $M = 1000$);

2. 16 sensors arranged in a grid, with readings at 5 time points, using 500 RFFs ($n = 80$, $M = 500$).

We generated three ground truth sources using length-scales $l = 5, 2$ and $1$. In each case, we used the adjoint method in the scenarios described to infer the posterior distribution of the source. Fig. 4 shows the ground truth generated with $l = 5$ and the inferred source in each scenario at a single time-slice. In this case both models perform similarly on visual inspection. The MSE between the inferred source and the ground truth is $0.004$ for scenario 1 ($n = 1000$, $M = 1000$), and $0.008$ in scenario 2 ($n = 80$, $M = 500$).

Fig. 5 shows the same information for the source generated with length-scale $l = 2$. In this case, we can visually see that the posterior inference is much more accurate in scenario 1. The MSEs are $0.07$ for scenario 1 and $0.68$ for scenario 2. Finally, in the case where $l = 1$ (see Fig. 6), visual inspection reveals that in scenario 2, the posterior mean bears little resemblance to the ground truth, whereas key features of the ground truth are visible in the posterior mean for scenario 1. This is reflected in the MSEs, which are $1.85$ for scenario 1 ($n = 1000$) and $2.55$ for scenario 2 ($n = 80$).

These results demonstrate the expected phenomena, namely that as the ratio of the length-scale to the grid size decreases, more features and observations are required to accurately infer the ground truth. Furthermore, in the short length-scale case the accuracy of inference is generally lower, as the source varies more between sensor locations than in the longer length-scale case. Additionally, in longer length-scale cases, fewer features and observations are required for high quality inference, thus enabling inference with less computational resource.

Finally, Fig. 7 shows the trace plot for the $q$ parameters for an implementation of the Metropolis Hastings algorithm in the case where $M = 10$ features are used. See the main text (Sect. 4.2) for details.

## Bayesian Optimisation Output

Although not our focus here, we note that we can infer the remaining parameters $\mathbf{p}_1$, $p_2$ (PDE) and $\tau^2, \lambda$ (GP hyperparameters) in a variety of ways. By way of illustration, here we use the GPyOpt (González & Dai, 2016) package to maximise the negative log-likelihood using the expected improvement acquisition function in a Bayesian optimization approach (Shahriari et al., 2015).

To perform inference for $\mathbf{p}_1, p_2, \tau^2, \lambda$, we wrote a function which, given a parameter array $\theta$, estimates the posterior predictive accuracy of our source posterior when using $\theta$ (here $\theta$ may contain a subset

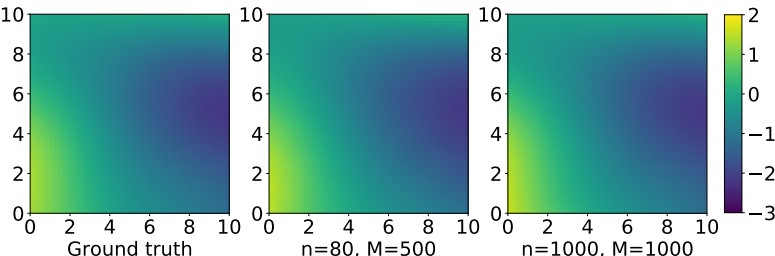

Figure 4: Ground truth and posterior mean of the source when using kernel length-scale, $l = 5$ (time-slice at $t = 5$). The left image shows the ground truth source, the middle image shows the posterior mean inferred using 80 observations and 500 features (MSE=0.008), the right shows the posterior mean inferred using 1000 observations and 1000 features (MSE= 0.004).

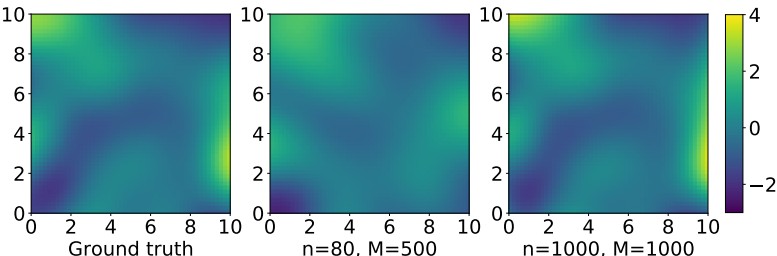

Figure 5: Ground truth and posterior mean of the source when using kernel length-scale, $l = 2$ (time-slice at $t = 5$). The left image shows the ground truth source, the middle image shows the posterior mean inferred using 80 observations and 500 features (MSE=0.68), the right shows the posterior mean inferred using 1000 observations and 1000 features (MSE=0.07).

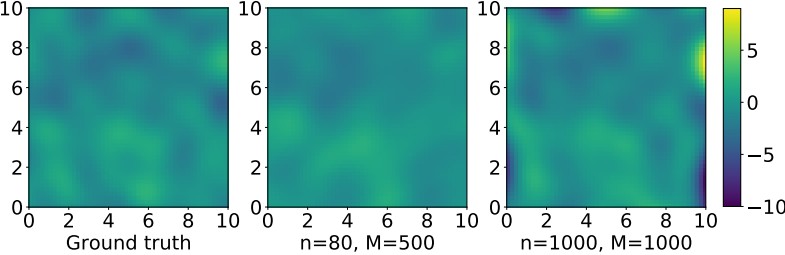

Figure 6: Ground truth and posterior mean of the source from with kernel length-scale, $l = 1$ (time-slice at $t = 5$). The left image shows the ground truth source, the middle image shows the posterior mean inferred using 80 observations and 500 features (MSE=1.85), the right shows the posterior mean inferred using 1000 observations and 1000 features (MSE=2.55).

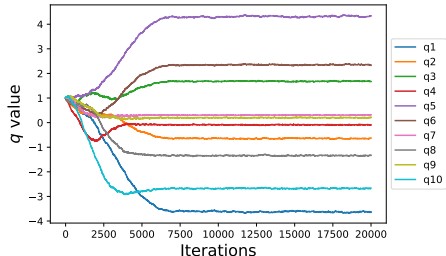

Figure 7: Trace plots for MCMC corresponding to Table 1

of the parameters). We do this by first simulating 100 realizations from the posterior mean of the source for a fixed parameter, i.e., $f_1, \ldots, f_{100} \sim p(f|z, \theta)$, and we push these through the PDE to get a posterior predictive sample of concentration fields $u_1, \ldots, u_{100} \sim p(u|z, \theta)$. The negative log-likelihood is then calculated between these and the training observations giving us a way to score parameter $\theta$. The negative log-likelihood function was used as the objective function in a Bayesian Optimisation routine González & Dai (2016). To test this approach we generated various ground truth source and solution fields using fixed values of $\theta$.

Fig. 8 shows the exploration and eventual convergence in a particular case where we used $\lambda = 2$ to generate a ground truth source. In this case, the maximum likelihood estimate of $\lambda$ was found to be $\hat{\lambda} = 2.52$ which the optimization found after 29 iterations. In a case where the true kernel length-scale and variance were both 2, and the wind-speed was 0.04, Bayesian optimisation found the maximum likelihood values of 1.24, 3.20 and 0.031 respectively after 20 iterations. Further work is needed to fully explore how to embed this approach into parameter estimation schemes, but we hope it gives some insight into how parameter estimation could be performed. Finally, note that the adjoint approach gives the possibility of estimating the gradient of the loss function. This would enable the adjoint approach to be embedded into gradient based inference algorithms such as the VAE and Hamiltonian Monte Carlo, hopefully accelerating inference.

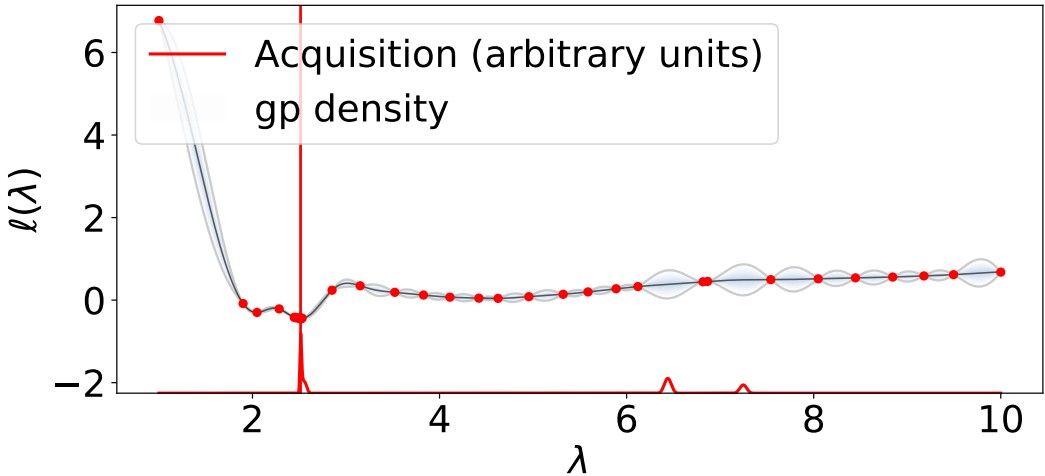

Figure 8: The output of the Bayesian Optimisation algorithm used to infer the value of the GP kernel length-scale, $\lambda$. In this case the true value of $\lambda$ is 2 and the algorithm found the minimum of the negative log-likelihood at $\lambda = 2.52$. This plot shows where the objective function was evaluated, and the posterior mean and variance at each point.

**PDE model applied to the Round Hill II dataset**

To test the approach using data from a physical experiment, we used the Round Hill II advection-diffusion experiment (Cramer & Record, 1957). In this study, researchers deployed 183 midget impingers for measuring sulphur dioxide in three partial concentric rings, 50m, 100m and 200m downwind from the release site, spanning $69°$. A constant source of sulphur dioxide (releasing approximately 5-10 $gs^{-1}$) was used over a ten minute period, during which the impingers took measurements of average concentration over the first 30 seconds, 3 minutes and 10 minutes. The average wind speed and direction was recorded ($2.14\ ms^{-1}$). We modelled this with our adjoint approach, as in section 4.2, over a $250m \times 250m$ domain spanning 13 minutes, using 10,000 random Fourier bases to approximate the Gaussian process forcing term. We tested two aspects of our model's capabilities. First: Source attribution. The model's mean source prediction was roughly flat except for a peak approximately 45m downwind of the true release site, see Figure 9. This discrepancy is expected as the true dataset contained a point source while our model had a GP prior (with EQ kernel and lengthscale of 10m) over the source. This leads to an inferred broader source, slightly closer to the ring of sensors. The second test was predicting the $SO_2$ concentration: We removed the middle (100m) ring of sensors from the training data, then tried to predict their measurements.

For comparison we used a Gaussian process with a length-scale of 30m (and 30s) to predict the concentration. We found it useful to threshold the concentrations to be non-negative for both methods. Our model performed considerably better than the GP model. For the three measurement periods the results were:

```
        Our Model    GP Model
 30s    14444         21385
180s     6628         12968
600s     4503          8490
```

[measurement units were $mg/m^3$, so these MSEs are in $(mg/m^3)^2$ ] See figure 10 for a comparison of the inferred concentrations between the two models. Figure 10b indicates that the Gaussian process generally overestimated the right hand side of the left out sensor array, whereas it can be seen from 10a that the adjoint method predicted the true high concentration area fairly well, with some smaller overprediction at the left and right hand sides of the array.

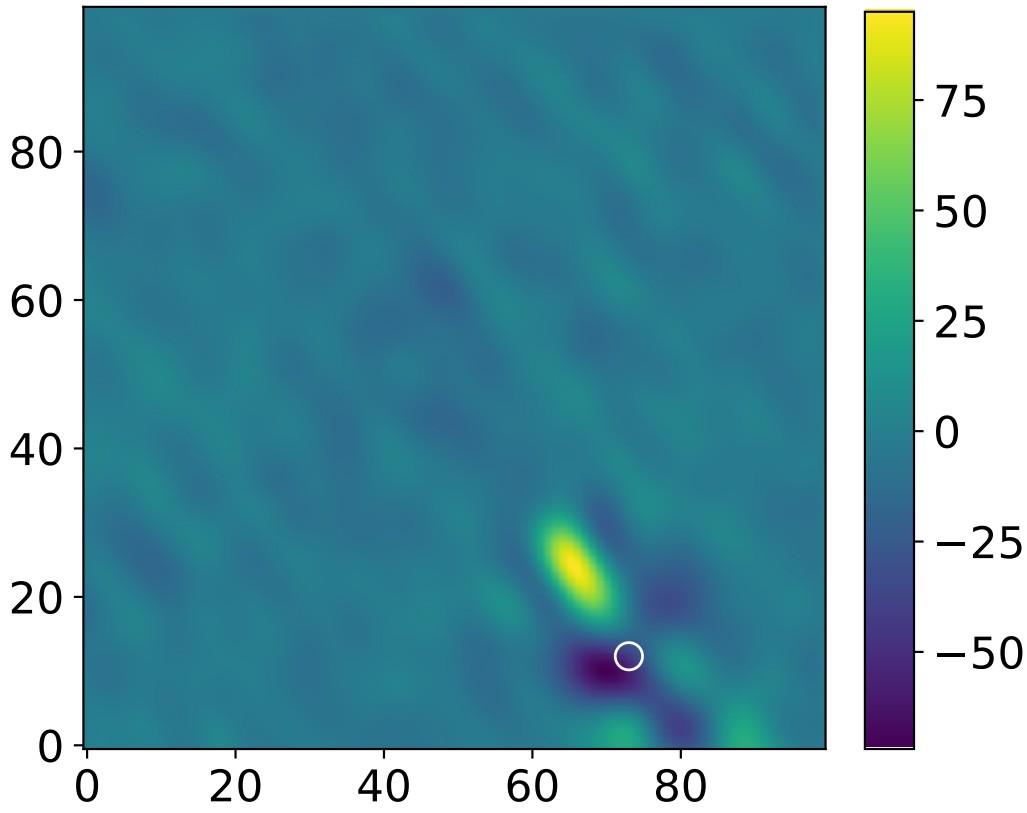

Figure 9: The mean inferred source in the Roundhill experiment at $t = 0$. The white circle indicates the true source location.

## Shift operators

Here we provide an example of the adjoint method applied to a non-differential operator: the shift operator. Consider the operator $\mathcal{L}_a : \mathbb{R} \to \mathbb{R}$ such that

$$\mathcal{L}_a(u(t)) = u(t + a). \tag{26}$$

This is the right shift operator. We can derive the adjoint of the right shift operator by taking the following inner product and using a change of variable (x=t+a).

$$\langle \mathcal{L}u, v \rangle = \int_{-\infty}^{\infty} u(t + a)v(t)dt = \int_{-\infty}^{\infty} u(x)v(x - a)dx \tag{27}$$

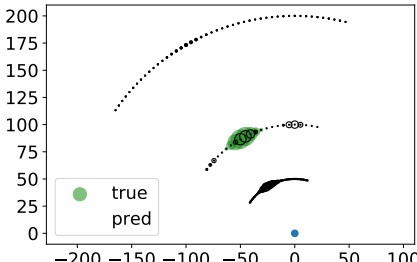
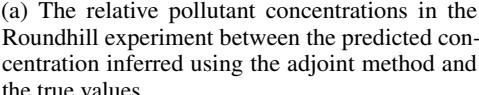

(a) The relative pollutant concentrations in the Roundhill experiment between the predicted concentration inferred using the adjoint method and the true values.

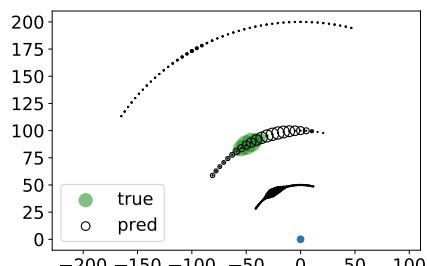

(b) The relative pollutant concentrations in the Roundhill experiment between the predicted concentration inferred using a Gaussian process and the true values.

Figure 10

and so the adjoint of the right shift operator is the left shift operator, $\mathcal{L}_a^* : \mathbb{R} \to \mathbb{R}$, where $\mathcal{L}_a^* v(t) = v(t-a)$. Having derived the adjoint of the right shift operator, it is possible to apply the adjoint method to an example system

$$\mathcal{L}_a u(t) = u(t+a) = f(t) \tag{28}$$

where $f(t)$ is an unknown forcing function and observations of $u$ are obtained as noisy averages over short time windows (see equation 16).

Figure 11 shows the inferred and true $f(t)$ and $u(t)$ of the system given in equation 28 with $a = 2$ and 20 observations evenly spaced between $t = 2$ and $t = 8$. Observations were generated with Gaussian noise, $\epsilon \sim N(0, 0.05)$. The output $u(t)$ is well predicted within the observation range with relatively high certainty, the prediction is uncertain outside of the observation range. The forcing function, $f$ is well predicted between $t = 0$ and $t = 6$, i.e. the observation range shifted left by $a = 2$. The MSE between the observations and the mean value of $u(t)$ inferred at the observations points is 0.003. For reference, the standard deviation of the observations is 1.001.

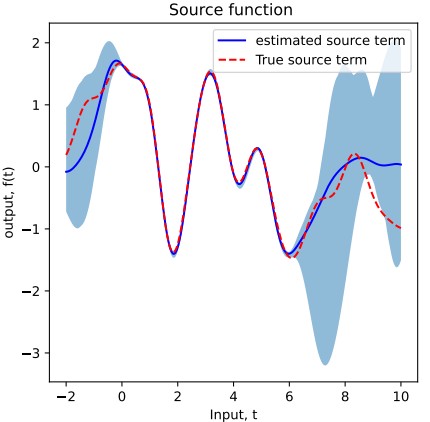

(a) Inferred and ground truth source in the shift operator system with $a = 2$

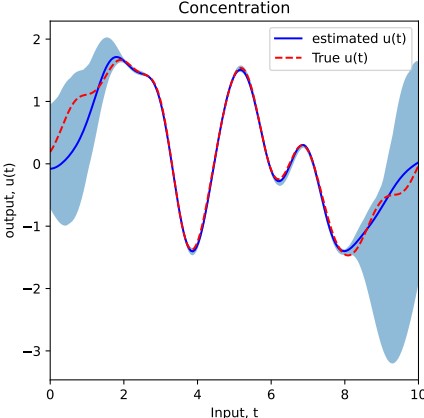

(b) Inferred and ground truth output, $u(t)$ in the shift operator system with $a = 2$

Figure 11

It seems evident that to predict the source and output more confidently over the entire real line would require observations over the entire real line. Furthermore, it does not seem possible to infer the shift parameter, $a$. For example, if the true shift parameter is $a^*$ and the the model used to infer the source assumes shift parameter $a$, the inferred source will simply be shifted left by $a$ and the quality of prediction of observation points would be indistinguishable (given a fixed basis for expansion of $f$).

## Supplementary References

Alvarez, M., Luengo, D., and Lawrence, N. D. Latent force models. In *Artificial Intelligence and Statistics*, pp. 9–16. PMLR, 2009.

Alvarez, M. A., Luengo, D., and Lawrence, N. D. Linear latent force models using Gaussian processes. *IEEE Transactions on Pattern Analysis and Machine Intelligence*, 35(11):2693–2705, nov 2013. ISSN 1939-3539. doi: 10.1109/TPAMI.2013.86.

Boyle, P. and Frean, M. Dependent Gaussian processes. In Saul, L., Weiss, Y., and Bottou, L. (eds.), *Advances in Neural Information Processing Systems*, volume 17. MIT Press, 2005. URL https://proceedings.neurips.cc/paper/2004/file/59eb5dd36914c29b299c84b7ddaf08ec-Paper.pdf.

Cole, K. Green's Function Library, 2000. URL http://www.engr.unl.edu/~glibrary/glibcontent/glibcontent.html.

Cramer, H. E. and Record, F. A. Field studies of atmospheric diffusion and the structure of turbulence. *American Industrial Hygiene Association Quarterly*, 18(2):126–131, 1957.

Estep, D. A short course on duality, adjoint operators, Green's functions, and a posteriori error analysis. *Lecture Notes*, 2004.

González, J. and Dai, Z. GPyOpt: A Bayesian optimization framework in python (BSD 3 revised license). http://github.com/SheffieldML/GPyOpt, 2016.

Guarnizo, C. and Álvarez, M. A. Impulse response estimation of linear time-invariant systems using convolved gaussian processes and laguerre functions. In Mendoza, M. and Velastín, S. (eds.), *Progress in Pattern Recognition, Image Analysis, Computer Vision, and Applications*, pp. 281–288, Cham, 2018. Springer International Publishing. ISBN 978-3-319-75193-1.

Guarnizo, C. and Alvarez, M. A. Fast kernel approximations for latent force models and convolved multiple-output Gaussian processes. In *Uncertainty in Artificial Intelligence*, pp. 835–844, 2018.

Higdon, D. Space and space-time modeling using process convolution. In *Quantitative methods for current environmental issues*, pp. 37–56. Springer, 2002.

Rasmussen, C. E. and Williams, C. K. I. *Gaussian processes for machine learning*. MIT press Cambridge, MA, 2006.

Shahriari, B., Swersky, K., Wang, Z., Adams, R. P., and De Freitas, N. Taking the human out of the loop: A review of bayesian optimization. *Proceedings of the IEEE*, 104(1):148–175, 2015.

Tobar, F., Bui, T. D., and Turner, R. E. Learning stationary time series using Gaussian processes with nonparametric kernels. In *Advances in Neural Information Processing Systems*, pp. 3501–3509, 2015.