# OpenReview forum: "Adjoint-aided inference of Gaussian process driven differential equations"
_NeurIPS.cc/2022/Conference — NeurIPS 2022 Accept_

### Official Review · Reviewer_mA14 · 2022-07-05

**Rating:** 4
**Confidence:** 3
**Soundness:** 2 fair
**Presentation:** 3 good
**Contribution:** 2 fair

**Summary:**

The authors developed a method to learn unknown forcing functions $f$ of linear systems $Lu=f$ from (affine) observations of $u$, also for PDEs. For finite dimensional learning, this is done symbolically by means of the adjoint. For infinite dimensional learning, a Gaussian process prior is made "finite dimensional" by means of a truncated basis expansion.

**Questions:**

**Questions**
- I might have failed to understand the explanation for the linear runtime when using a GP and would like the authors to elaborate on this. Is the GP never calculated, but rather $z$ is estimated and using the adjoint, the resulting $f$ can be estimated?
- How general is the suggested method really? Did you try shift operators?
- Is your method applicable to the case when u is not given via observations but as a GP itself? I.e., can you solve linear operator equations for GPs?

**Suggestions**
- A quick reminder in the beginning of 3.1 that $L:\mathcal{U} \to \mathcal{V}$, might improve the flow of reading
- Inconsistencies in how the authors write equations
	- line 172 Eq. 13) and line 179 (Eq. 14)
- The typesetting of the small brackets in the left part of equation (11) is awkward.

**Limitations:**

The authors have not brought up limitations of their work. I do not see a necessity, apart from the above weaknesses.

**Strengths And Weaknesses:**

**Strengths**
The authors leverage non-trivial ideas from functional analysis to construct better models in a relevant and significant area of research.
The technique is not original in mathematics, but it is a non-trivial extension to machine learning.
Comparison to the state of the art is clearly given (even though, of course it could be longer, but the number of pages is limited).
The language is clear and readable.

**Weaknesses**
1) After reading the paper, there remains a big gap/error in the functional analysis part of the paper. (This might just be a gap/error in the theoretical justification of the method. I do not claim that the method is wrong.)
   Under one viewpoint, three kinds of function spaces appear:
     (i) Hilbert space. They are used a lot, e.g. to compute adjoints. And the realizations of a GP with EQ kernel lie in such spaces (non-trivially, see arXiv:1807.02582).
	 (ii) Banach spaces. I do not know why these spaces appear in the paper.
	 (iii) Frechet spaces, which is the class of spaces your GP-realizations with the EQ kernel have a measure on (see arXiv:2205.03185). (These realizations lie in the space of smooth functions, which is not Banach w.r.t. its usual topology.)
   Of course, there are Banach spaces suitable for GPs (probably for the Matern kernel, though I do not have a reference).
   Furthermore, the computation of the adjoint as elements in a Hilbert space (using the Riesz representation theorem) depends on the choice of the scalar product. In the example in Section 4.1 such a scalar product is chosen. Hence, the result depends on this choice. This choice (and hence the adjoint, and hence the entire computation) are arbitrary. Even as we have learned in arXiv:1807.02582, there seems to be no canonical RKHS, which contains the GP realizations, and hence no canonical choice of a scalar product. I would assume that the choice of the topology to define the adjoint is clearly justified.

   Perhaps I am missing a trivial point here, but I did not find this point in the paper. A summary of the paper according to my feeling is: "Talk about Hilbert and Banach Spaces. Some math magic happens, which we do not explain. This justifies our method." Again, I am not claiming that the method is wrong. But please either explain this math magic or do not try to sound mathmatical.
   If the authors can clarify this point convincingly in a response, I am willing to drastically improve my scores, as otherwise (apart from the following points (2) and (3), which are not too damning) the paper is rather good.

2) In my opinion, the authors did not clearly show the benefit of their method experimentally. Neither a comparison with a standard GP was made, nor did the authors compare themselves to related work that might have tackled the same task. Adding this to the experiments would greatly improve the papers value.
3) Experiments on real data would have helped. When dealing with differential equations, synthetic data is usually to "nice".

---

> ### Author Response · Authors · 2022-08-02
> **Response to weaknesses 1 and 2**
>
> We would like to begin by thanking the reviewer for their time and thoughtful
> comments. Below we have attempted to address their questions and concerns.
>
> Weaknesses:
>
> 1. “A big gap/error in the functional analysis part of the paper. “
>
>
>
> Thank you for pointing out the lack of clarity on this - we agree this could be clearer in the paper and will modify accordingly. We think the root of the confusion is thinking about the problem from the wrong direction: rather than thinking about what space GP samples lie in, we should instead think about what space solutions to the linear system lie in.
>
>
>
> Which function space are we working in?
>
>
>
> Let’s focus on the more complex PDE case. When solving a PDE, we first select a function space in which to work - we then seek solutions within this space of functions. The typical choice is some Sobolev space with sufficiently many derivatives for the given system, i.e., a specific Banach space. This was our reason for presenting the adjoint system in its general Banach formulation.
>
>
>
> Once this function space is fixed, we then consider GP forcing functions projected onto this space. This works because i) we solve the systems numerically by discretising space and time, so that in practice everything is finite dimensional; and ii) we work with a truncated GP expansion, with basis vectors that are well behaved elements belonging to any function spaces we are likely to care about in practice.
>
>
>
> The adjoint depends on the chosen inner product, and the choice is arbitrary?
>
>
>
> Yes. When we set up the PDE and observation operator, we choose a function space to work in. This choice is made by the modeller and is a matter of scientific judgement. Once the choice is made, it then defines the function space for the solution of the adjoint systems.
>
>
>
> In the specific examples considered, we used the standard inner-product typically used for Lebesgue spaces. This is used in the two adjoint derivations (the displayed equation after line 181 for the ODE, and Eq. 21 in the supplement for the PDE), but is not stated explicitly. We agree this could be more clearly stated as a choice specific to each example, and will make this clearer in the revised version of the paper.
>
> 2: Comparison to methods in the related work:
>
> We conducted a comparison between the adjoint method, the Green's function method and a classical Gaussian process on the ordinary differential equation example. Observations were taken at 20 time points over $t\in[0,10]$ with a grid resolution of 200.
>
> - The Gaussian process had a mean squared error of 0.0055 between the true output and the inferred output.
> - The Green's function method (as in Alvarez et al., 2009) had an MSE of 0.0051 for the output error and 0.0860 for the source error.
> - The Green's function method with random Fourier features (as in Guarnizo and  Álvarez, 2018) achieved the following MSEs:
>
>     - 20 features: Source MSE of 0.099 and output MSE of 0.0058
>
>     - 200 features: Source MSE of 0.0927 and output MSE of 0.0055.
>
>     - 500 features: Source MSE of 0.0856 and output MSE 0f 0.0052.
>
>     - 2000 features: Source MSE of 0.0861 and outpute MSE of 0.0051.
>
> - The adjoint method with M=200 random Fourier features had an MSE of 0.0056 between the ground truth concentration and the inferred concentration and an MSE of 0.079 between the ground truth and inferred sources.
>
> All three methods achieve a similar quality of inference over the system output. For larger numbers of features ($M\sim200$) the adjoint method and the GP predicted the system response with similar accuracy. It should be noted that by using a classical GP approach it is not possible to infer the unknown forcing function, $f$, which is one of the key advantages of the adjoint method. The Green's function method also performs to a similar level of accuracy as the adjoint method. We are happy to include this comparison in the revised draft of the paper.

---

> > ### Author Response · Authors · 2022-08-02
> > **Response to weakness 3**
> >
> > Weaknesses:
> >
> > 3. There should be more experiments:
> >
> > To test the approach using data from a physical experiment, we used the Round Hill II advection/diffusion experiment\footnote{Cramer, H. E., and F. A. Record. "Field studies of atmospheric diffusion and the structure of turbulence." American Industrial Hygiene Association Quarterly 18.2 (1957): 126-131.}. In this study, researchers deployed 183 midget impingers for measuring sulphur dioxide in three partial concentric rings, 50m, 100m and 200m downwind from the release site, spanning $69^\circ$. A constant source of sulphur dioxide (releasing approximately 5-10 $g s^{-1}$) was used over a ten minute period, during which the impingers took measurements of average concentration over the first 30 seconds, 3 minutes and 10 minutes. The average wind speed and direction was recorded (2.14 $m s^{-1}$). We modelled this with our adjoint approach over a $250m \times 250m$ domain spanning 13 minutes, using 10,000 RFBs. We tested two aspects of our model's capabilities. First: Source attribution. The model's mean source prediction was roughly flat except for a peak approximately 45m downwind of the true release site. This discrepancy is expected as the true dataset contained a point source while our model had a GP prior (with EQ kernel and lengthscale of 10m) over the source. This leads to an inferred broader source, slightly closer to the ring of sensors. The second test was predicting the SO${}_2$ concentration: We removed the middle (100m) ring of sensors from the training data, then tried to predict their measurements. For comparison we used a Gaussian process with a length-scale of 30m (and 30s) to predict the concentration. We found it useful to threshold the concentrations to be non-negative for both methods. Our model performed considerably better than the GP model. For the three measurement periods the results were:
> >
> >             Our Model    GP Model
> >     30s   20180        21749
> >     180s   3766        13086
> >     600s   3946         8506
> >
> > [measurement units were $mg/m^3$, so these MSEs are in $(mg/m^3)^2$ ]
> >
> > We will include these results, and associated figures in the final paper. It is likely that the prediction for the held-out 30s observations is poor as there are not observations upwind and close enough, for advection to allow these early test points to be estimated. By 180s however, air which has visited the training points has also visited the test points, and vice-versa, allowing more accurate estimates.

---

> > > ### Author Response · Authors · 2022-08-02
> > > **Response to questions 1 to 4 and suggestions**
> > >
> > > Questions:
> > >
> > > 1. Is the GP calculated?
> > >
> > >     The GP is never explicitly calculated, we use the adjoint method to generate a distribution over $q$, which is then used to calculate the source as in equation 8. We realise that section 3.3 was unclear on this procedure and aim to rewrite it to improve the clarity in the revised draft of the paper.
> > >
> > > 2. Does this method work for shift operators?
> > >
> > >     Though we had not tried the method on shift operators, we have attempted applying our method to them in light of this question. We establish the linear system $L_a(u(t,x,y))=u(t+a,x,y)=f$, where $L_a$ is the right shift operator. Using the fact that the adjoint of the right shift operator is the left shift operator ($L_a^*(v(t,x,y)=v(t-a,x,y)$) and applying the adjoint method, we have been able to infer the forcing function $f$ and system response $u$ in the system $L_a(u(t,x,y))=u(t+a,x,y)=f$ on a bounded domain. We generated a test source as in the Advection Diffusion example in the paper and the MSE between the ground truth forcing function and the inferred forcing function was 0.48 and the MSE between the true concentration and the inferred concentration was 0.40. The data variance was 1.59, for comparison. We would be happy to include this experiment and associated figures in the revised supplementary material.
> > >
> > > 3. Is your method applicable to the case when u is not given via observations but as a GP itself? Can you solve linear operator equations for GPs?
> > >
> > > At the time of writing we are unsure of whether it is possible to apply our method to the case when u is not given via observations but as a GP itself. This is a fascinating idea and one that we will consider for future work. Now, to answer your more general question on solving linear operator equations for GPs, this is indeed possible and researchers have done it in the past. See for example López-Lopera et al. (2019). Since u is a GP, it means it is defined by a mean function and a covariance function. We can then solve for f and compute the mean and covariance functions for f in terms of the mean and covariance functions for u. This operation will imply taking derivatives wrt the input dimensions for the covariance function of u. We are happy to clarify further if necessary.
> > >
> > > Suggestions:
> > >
> > > We are happy to adjust the manuscript to address the problems noted.
> > >
> > > References:
> > >
> > > A. F. López-Lopera et al., Physically-inspired Gaussian processes for transcriptional regulation in Drosophila melanogaster, IEEE/ACM Transactions on Computational Biology and Bioinformatics, https://arxiv.org/abs/1808.10026, 2019.

---

> > ### Comment · Reviewer_mA14 · 2022-08-03
> > **Response to response to weaknesses 1**
> >
> > Thx for your long reply to my review. The points in your answer are mostly reasonable. I would like to note that there does not seem to be a major improvement in in the state of the art w.r.t. accuracy.
> >
> > However, I need to ask more regarding the functional analysis, before I can think about improving my score.
> > Now you have three main function spaces:
> > 1) the Hilbert space for computing the adjoint,
> > 2) the Sobolev (Banach) space for the solutions, and
> > 3) the (Frechet for SE covariance) for the GP realizations.
> > Perhaps there is even a fourth function space:
> > 4) the right hand side (forcing function) might lie in its own function space.
> >
> > This setup automatically yields questions:
> >
> > - i) Do you need or consider embedding theorems from any of the function spaces into each other? Are these embeddings continuous or surjective?
> > - ii) Are there other compatibility conditions between these spaces? (E.g. can you use any Hilbert space structure to compute the adjoint?)
> > - iii) Do you need or consider continuity of your operators on these spaces? Are the operators continuous or surjective?
> > - iv) What requirements do you have w.r.t. the right hand side being in the image of the operator?
> > - v) Does this depend on the operator, e.g. its order? Does this change for certain classes of operators? Does this change for ODEs?
> > - vi) Do you have examples of quadruples of classes of function spaces where all of this works?
> >
> > Since you answers spawn new questions from my side, I would like to see the revised formulations in the paper. You are working in a highly non-trivial area of functional analysis and PDEs (and, to be honest, this is far from my area of expertise, and it seems even farther away from the area of expertiese of the other reviewers), where I just cannot blindly sign off on changes.

---

> > > ### Author Response · Authors · 2022-08-04
> > > **Response to your further questions**
> > >
> > > Many thanks. Regarding accuracy: We agree this approach will not be more accurate, as the underlying model is the same as is solved using finite differences and MCMC, or a Green's function approach. The purpose of this paper is to show an alternative approach to solving the model that takes a similar amount of time to the Green's function approach (i.e. much faster than an MCMC approach), but does not require finding the relevant Green's function for a given ODE/PDE and boundary conditions (which can often be quite challenging and cumbersome) and can be applied to non-differential linear operators where a Green's function does not exist.
> > >
> > > The further points you raise are interesting and raise deep questions of functional analysis. These questions deserve careful treatment by analysts, but that is best done in a suitable mathematics journal. We do not have complete answers to your questions, but as a partial response, we would like to make two observations. Our first is that in practice we are forced to use numerical methods, reducing the system to finite dimensional linear algebra, in which case these issues are no longer of concern. Secondly, we sketch an approach which could lead to a resolution of some of these points if developed fully - we acknowledge this is far from a complete answer to your questions. We would argue that (as you say) these are deep functional analysis questions, and that a NeurIPS conference paper would not be the place to develop such theory. We also note that most of the issues you raise here are also potential problems for any latent force model. What we have done is to approximate functions living in a range of spaces, and have produced a method that is equivalent to the state-of-the-art in terms of accuracy and computational efficiency. We believe the paper will be of interest to the community, and that our paper will be a useful addition to the literature in this area.
> > >
> > > We'll now expand on each of these arguments.
> > >
> > > Firstly, we want to stress that functional analysis issues are of relatively little concern in practice. Why so? Because we cannot solve the system mathematically, and so resort to numerical methods to find approximate solutions. We use a finite difference approach, which essentially reduces everything to a finite dimensional Euclidean space. The solution $u$ and forcing $f$ are essentially considered to be vectors in $\mathbb{R}^D$, let's say $\mathbf{u}, \mathbf{f} \in \mathbb{R}^D$ to make specific notation for the finite dimensional versions. Here, $D=NT$ where $N$ is the number of points in the spatial discretization, and $T$ the number of points in the temporal discretization. The linear operator $\mathcal{L}$ can then be considered as a $D \times D$ matrix, let's say $\bf{L}$. The observation operator $h$ is a $n \times D$ matrix (where $n$ is the number of observations), so that the observations are $\bf{y} = \bf{h} \bf{u}+\bf{e}$. The adjoint operator is also a $D \times D$ matrix, and is equal to the transpose of $\bf{L}$, i.e., $\bf{L}^* = \bf{L}^\top$. The GP approximation to $\bf{f}$ is now a plain multivariate Gaussian distribution, and we truncate this as
> > > $\mathbf{f} = \sum_{i=1}^M \lambda_i z_i \mathbf{\phi}_i$
> > > where $\mathbf{\phi}_i$ are vectors in $\mathbb{R}^D$ formed by evaluating the RFFs at the grid points.  Thus we're left with a set of linear finite dimensional equations, which our paper shows how to solve. These are approximate solutions that will become more accurate as $N$, $T$ and $M$ increase. Given any finite choice for these values, our method gives an exact solution to the approximate problem (in contrast, MCMC would give an approximate solution to the approximate problem). However, there is only minor interest in the exact solution in practical settings, as the mathematical equations themselves are just approximations to reality. Untangling this mess of approximations is largely a matter of empirical science in any given situation - we try to see if adopting models and numerical treatments of models leads to good predictions and inferences, and then seek to improve the model,

---

> > > > ### Author Response · Authors · 2022-08-04
> > > > **Response to your further questions (continued)**
> > > >
> > > > Secondly, we give a preliminary sketch of an approach that could lead to an answer to some of your functional analysis questions. There are 3 spaces:
> > > > $\mathcal{U}$ the solution space, with $u\in \mathcal{U}$;
> > > > $\mathcal{V}$ the space in which $\mathcal{L}u$ belongs;
> > > > $\mathcal{W}$ the space containing the GP samples, $f \in \mathcal{W}$;
> > > > $\mathbb{R}^n$ the space the observations lie in.
> > > >
> > > > The linear operator maps
> > > > $$\mathcal{L}: \mathcal{U} \rightarrow \mathcal{V}$$
> > > > The observation operator maps
> > > > $$h: \mathcal{U} \rightarrow \mathbb{R}^n$$
> > > > The adjoint operator maps
> > > > $$\mathcal{L^*}: \mathcal{V}^* \rightarrow \mathcal{U}^*$$
> > > > where $\mathcal{U}^*$ denotes the continuous dual space of $\mathcal{U}$ etc.
> > > > If $\mathcal{W}$ (the space of GP samples) can be embedded in $\mathcal{V}$ (the codomain of $\mathcal{L}$),  the effect is to restrict the space of possible solutions, i.e., instead of finding solutions in $\mathcal{U}$, we find solutions in $\mathcal{U}' = \{u\in \mathcal{U}  : \mathcal{L} u \in \mathcal{W}\}$. For the model to produce meaningful inferences, we would want the embedding to be continuous, but it need not be surjective or dense.
> > > >
> > > > So what are these spaces in practice? Sticking with the PDE example, let's just consider the case where $V = W^{k,2}=H^k$ - the Sobolev space of functions on $\mathcal{X}\times \mathcal{T}$ (the spatio-temporal domain) with weak derivatives upto order $k$ with the $L_2$ norm. These are Hilbert spaces, with inner-product formed by sums of the $L_2$ inner-product. For $\mathcal{W}$, consider the space containing GP samples for the SE kernel. Is $W$ embedded in $V$? Certainly the RKHS corresponding to the SE kernel is embedded in $H^k$, but what about the samples? We haven't been able to prove this given the short rebuttal window, but it seems likely that a suitable continuous embedding exists. We're comforted by the following statement in the paper 2018 by Kanagawa et al (arxiv:1807.02582) comparing the sample space of a SE GP, with the RKHS corresponding to the SE kernel:
> > > >
> > > > `Therefore in practice one should not worry too much about the fact that a GP sample path almost surely falls outside the RKHS $H_{k_\gamma}$, because it nevertheless lies almost surely on the “infinitesimally small shell" surrounding $H_{k_\gamma}$'
> > > >
> > > > In other words, the sample space is not much larger than the well-behaved RKHS embedded in $H^k$, and so it seems likely such an embedding does exist. However, if we think specifically about the random Fourier feature (RFF) approximation we use, this essentially has the effect of projecting samples from the GP back into the RKHS corresponding to the SE kernel, and so in practice, for finite choices of $M$, there is no mismatch between function spaces.
> > > >
> > > > We hope this goes some way to satisfying your concerns. The discussion has helped clarify our thinking on these issues and we will update the paper accordingly to clarify our viewpoint, and to discuss the potential theoretical issues.

---

> > > > > ### Comment · Reviewer_mA14 · 2022-08-04
> > > > > **Hopefully final response from me**
> > > > >
> > > > > Thx for your response.
> > > > >
> > > > > Some points:
> > > > > - I think we are in agreement that this entire functional analysis cannot be developed in a NeurIPS paper. Perhaps I would have complained a lot less, if there wasn't such a strong focus on Banach spaces and Hilbert spaces in your paper, and then you do not think it through to the end.
> > > > > - I would advise you to say less about functional analysis in your main paper, but give details (as above) in your appendix.
> > > > > - I will raise my score to 4, which is still a (borderline) reject, since there is much to change in your paper. Seeing the other reviews, you might have a chance to get accepted. And I think it should be an editorial descision whether to accept your paper, since (i) the ideas are good and certainly worthy of NeurIPS and (ii) there are several technical problems that the editors need to trust you to get rid off in the camera ready version. I personally would still reject, but factoring in your replay, I could now understand a descision in the other direction.
> > > > >
> > > > > One comment:
> > > > > Proposition 1 in arXiv:2205.03185 embeds the realizations of the SE-kernel into the set of smooth functions. This might be an alternative embedding theorem to the one in arxiv:1807.02582. Not sure which (if any) of these two helps you.

---

> > > > > > ### Author Response · Authors · 2022-08-05
> > > > > > **Thank you for revising your score and a further note**
> > > > > >
> > > > > > Thanks for clarifying, and thank you for noting that the ideas in the paper 'are good and certainly worthy of NeurIPS'. We think NeurIPS is the ideal venue for interesting new ideas and we hope the area chairs will take this into consideration.
> > > > > >
> > > > > > We don't think your comment about there being a 'strong focus on Banach and Hilbert spaces' is fair - would you reconsider this? Our paper is about inference for forcing functions for linear operators. A linear operator maps from one space to another, and it is surely necessary for us to define what type of spaces our methods apply to. The most general type of space for which we can define adjoint operators, are Banach spaces. These aren't exotic or mathematically niche spaces - they are vector spaces with a distance (norm).
> > > > > > In the paper, the only functional analysis is that we define what an adjoint operator is, and we show the specialization to the more familiar Hilbert space setting (vector space with angle and distance). Adjoint operators may not be widely understood in the NeurIPS community, but they are key to our paper, and so defining them in the paper seems essential? We don't think it is fair to describe this as being overly mathematical, or as trying to 'sound mathematical' - what would the alternative be? If we presenting everything in a finite dimensional Hilbert space (ie. matrix algebra) and just used the matrix transpose as the adjoint, then it would make the application to the differential equation setting much messier to explain, and the simplicity and wide applicability of the method would be lost.
> > > > > >
> > > > > > You say you would have complained less and given us a drastically improved score if we'd not attempted to be mathematical, please can we ask you to think again about whether a less 'mathematical' presentation, whatever that would be, would add to the clarity of the paper?
> > > > > >
> > > > > > Re the 'technical problems', could you respond to our comment about there being nothing of concern in the practical setting where we use a truncated basis expansion of GPs, and use an numerical approximation (finite difference in our case) to solve the differential equations? In the continuous case, we can see there are some abstract mathematical details that may need clarification, and we're glad that you think our sketch of how these would be fixed sounds intuitively plausible. We're also glad you do not think NeurIPS is the appropriate place to develop such detail.

---

### Official Review · Reviewer_jfqV · 2022-07-06

**Rating:** 6
**Confidence:** 3
**Soundness:** 3 good
**Presentation:** 3 good
**Contribution:** 3 good

**Summary:**

This paper is concerned with the inverse problem of estimating the forcing function of f of a linear system Lu=f, with affine linear observations, where the prior on f is a Gaussian process. The authors introduce a trick. They propose to use the adjoint linear system L* v = h in order to rewrite the observation vector as a linear transformation of the parameters of f (plus noise). This of course assumes that f can be decomposed into such an expansion. If f is modeled as a GP, then a truncated basis expansion can play this role - such that the posterior can be inferred.
This trick has the advantage of turning the model into a standard linear model. Now, standard results can be used to compute least squares (ML) or Bayesian estimators. This is advantageous because these estimators are now only simple linear algebra. The authors derive the trick for Hilbert spaces, but claim that it can be executed for more general Banach spaces.
In the experiments, this approach is spelled out both for ODEs and PDEs. A real time cost analysis is given, comparing with MCMC. Which approach is best, will depend on n. The authors discuss that, if n is not huge, their adjoint approach could outperform MCMC.

**Questions:**

1. Table 1: It is curious that there is no difference between MCMC and Adjoint in this data. Why is that?
2. The linear-parameter assumption of Eq. (8) is common for ODE inverse problems. See e.g. Assumption 1 in https://arxiv.org/pdf/2002.09301.pdf and Eq. (10) in https://proceedings.neurips.cc/paper/2017/file/e71e5cd119bbc5797164fb0cd7fd94a4-Paper.pdf . Maybe comment on that? In any case, it would be good for the reader to know how much of a restriction that assumption entails. What else does the existing literature propose to do with such an assumption?
3. Lines 141-151: Why is the solution to the adjoint equation v_i not part of the ML and Bayesian estimators.

**Limitations:**

There are no limitations and potential negative social consequences.

**Strengths And Weaknesses:**

Strength:
1. The paper is easy to read and straightforward. The authors do a good job of highlighting the core of their paper -- how the adjoint trick turns the system into a standard linear model, solvable by linear algebra. The derivation of the adjoint equation reads quite nicely.
2. The literature review is comprehensive.
3. The experiments are instructive and interesting.
4. The discussion is honest and transparent over the trade-offs in the variable n, compared to MCMC.
5. The method is novel, as far as I know.
6. The work is significant because (Bayesian) inverse problems are of great interest to the (probabilistic) ML community.

Weaknesses:
1. A few more experiments could have been given. Right now there is just one ODE and one PDE. More experiments would seem appropriate.
2. The authors compare with none of the methods cited in the Related Work section. It would be great to compare with some of the most recent methods.
3. Comments on the Computational Complexity are spread out over the paper. It would be good to write one section where all of these issues are discussed.
4. In line 152, the authors claim that their trick extends to "more general Banach spaces". I don't understand how. More justification for this claim should be given; perhaps in an Appendix.
5. Section 3.3. is a bit difficult to contextualize. I assume it shows how Bayesian inference of f works, when it is a GP. The crucial bit seems to be that the GP prior can be expressed by a truncated random series, similar to Eq. (8). If so, this connection should be more explicitly made - to improve the clarity of the writing.

---

> ### Author Response · Authors · 2022-08-02
> **Response to weakness 1**
>
> We would like to begin by thanking the reviewer for their time and thoughtful
> comments. Below we have attempted to address their questions and concerns.
>
> Weaknesses:
>
> 1. There should be more experiments:
>
> To test the approach using data from a physical experiment, we used the Round Hill II advection/diffusion experiment\footnote{Cramer, H. E., and F. A. Record. "Field studies of atmospheric diffusion and the structure of turbulence." American Industrial Hygiene Association Quarterly 18.2 (1957): 126-131.}. In this study, researchers deployed 183 midget impingers for measuring sulphur dioxide in three partial concentric rings, 50m, 100m and 200m downwind from the release site, spanning $69^\circ$. A constant source of sulphur dioxide (releasing approximately 5-10 $g s^{-1}$) was used over a ten minute period, during which the impingers took measurements of average concentration over the first 30 seconds, 3 minutes and 10 minutes. The average wind speed and direction was recorded (2.14 $m s^{-1}$). We modelled this with our adjoint approach over a $250m \times 250m$ domain spanning 13 minutes, using 10,000 RFBs. We tested two aspects of our model's capabilities. First: Source attribution. The model's mean source prediction was roughly flat except for a peak approximately 45m downwind of the true release site. This discrepancy is expected as the true dataset contained a point source while our model had a GP prior (with EQ kernel and lengthscale of 10m) over the source. This leads to an inferred broader source, slightly closer to the ring of sensors. The second test was predicting the SO${}_2$ concentration: We removed the middle (100m) ring of sensors from the training data, then tried to predict their measurements. For comparison we used a Gaussian process with a length-scale of 30m (and 30s) to predict the concentration. We found it useful to threshold the concentrations to be non-negative for both methods. Our model performed considerably better than the GP model. For the three measurement periods the results were:
>
>             Our Model    GP Model
>     30s   20180        21749
>     180s   3766        13086
>     600s   3946         8506
>
> [measurement units were $mg/m^3$, so these MSEs are in $(mg/m^3)^2$ ]
>
> We will include these results, and associated figures in the final paper. It is likely that the prediction for the held-out 30s observations is poor as there are not observations upwind and close enough, for advection to allow these early test points to be estimated. By 180s however, air which has visited the training points has also visited the test points, and vice-versa, allowing more accurate estimates.

---

> > ### Author Response · Authors · 2022-08-02
> > **Response to weaknesses 2 to 5 and questions 1 to 3**
> >
> > Weaknesses
> >
> > 2: Comparison to methods in the related work:
> >
> > We conducted a comparison between the adjoint method, the Green's function method and a classical Gaussian process on the ordinary differential equation example. Observations were taken at 20 time points over $t\in[0,10]$ with a grid resolution of 200.
> >
> > - The Gaussian process had a mean squared error of 0.0055 between the true output and the inferred output.
> > - The Green's function method (as in Alvarez et al., 2009) had an MSE of 0.0051 for the output error and 0.0860 for the source error.
> > - The Green's function method with random Fourier features (as in Guarnizo and  Álvarez, 2018) achieved the following MSEs:
> >
> >     - 20 features: Source MSE of 0.099 and output MSE of 0.0058
> >
> >     - 200 features: Source MSE of 0.0927 and output MSE of 0.0055.
> >
> >     - 500 features: Source MSE of 0.0856 and output MSE 0f 0.0052.
> >
> >     - 2000 features: Source MSE of 0.0861 and outpute MSE of 0.0051.
> >
> > - The adjoint method with M=200 random Fourier features had an MSE of 0.0056 between the ground truth concentration and the inferred concentration and an MSE of 0.079 between the ground truth and inferred sources.
> >
> > All three methods achieve a similar quality of inference over the system output. For larger numbers of features ($M\sim200$) the adjoint method and the GP predicted the system response with similar accuracy. It should be noted that by using a classical GP approach it is not possible to infer the unknown forcing function, $f$, which is one of the key advantages of the adjoint method. The Green's function method also performs to a similar level of accuracy as the adjoint method. We are happy to include this comparison in the revised draft of the paper.
> >
> > 3: The discussion of the computational complexity is too dispersed:
> >
> > We agree that the commentary on computational complexity is too dispersed throughout the paper. We will consolidate the discussion (including footnotes 1 and 2 on page 9) into a new section 3.4, immediately before section 4.
> >
> > 4: How does this method extend to more general Banach spaces:
> >
> > Thank you for highlighting our lack of clarity here. As in the examples given we concentrate on Lebesgue spaces with a standard inner product and the extension of our method to more general Banach spaces is not the key focus of this paper, to reduce confusion we are happy to remove this comment in the revised draft of the paper.
> >
> > 5: Section 3.3 is difficult to contextualise:
> >
> > We see that we were unclear here and should have stated explicitly that by using the basis expansion it is possible to write $f$ as a linear combination of $q$ and the basis vectors $\phi_i$. Section 3.3 will be re-written to clarify the connection between the linear forcing term assumption and the truncated Gaussian Process.
> >
> > Questions:
> >
> > 1. Why is there no difference between the MCMC and adjoint in Table 1?
> >
> > In this case both the adjoint method and the MCMC method converged to the same distribution over $q$ (to 2 decimal places), this indicates that for a given set of basis vectors, the adjoint method and MCMC method should provide similar results (given that the MCMC converges).
> >
> > 2. Comment on the linear parameter assumption:
> >
> > Equation 8 does not correspond to a linear parameter assumption as in the two references cited. In both Gorbach et al. and Kersting et al. the assumption is on the parameters multiplying the state variable (i.e. in those cases $\dot{u}=f(u,\theta)$) whereas in the method we present, $f$ does not have a $u$ dependence. We do not believe the requirement that $f$ be expressed as a linear combination of functions to be particularly limiting. However, our method is necessarily limited by the requirement that the operator $L$ be linear. Which means that our method cannot be applied to commonly used models in various disciplines (i.e. SIR models) without a linearisation step. We are happy to add a comment to this effect.
> >
> > 3. Why is the solution to the adjoint not in ML and Bayesian estimators?
> >
> > We apologise for the lack of clarity here; the solutions to the adjoint equation, $v_i$, are in $\Phi$ (see equation 9), $\Phi$ then appears in the posterior mean and covariance shown in equation 11. We will add a short comment to make this clearer in the revised version of the paper.
> >
> >
> >
> > References:
> >
> > Álvarez, Mauricio, David Luengo, and Neil D. Lawrence. "Latent force models." Artificial Intelligence and Statistics. PMLR, 2009.
> >
> > Guarnizo, Cristian, and Mauricio A. Álvarez. "Fast kernel approximations for latent force models and convolved multiple-output Gaussian processes." arXiv preprint arXiv:1805.07460 (2018).

---

> > > ### Comment · Reviewer_jfqV · 2022-08-03
> > > **Thank you for these nice replies!**
> > >
> > > Thanks, these replies are very nice and convincing! Thank you in particular for including the additional experiments in the revised version!
> > >
> > > I am contented by your reply! I am in favor of accepting this paper and will raise my score by 1.

---

### Official Review · Reviewer_bJxr · 2022-07-09

**Rating:** 5
**Confidence:** 3
**Soundness:** 3 good
**Presentation:** 3 good
**Contribution:** 2 fair

**Summary:**

The paper discusses a method that infers the forcing function $f$ of the differential equation $L u = f$ from linear observations of $u$, $z = H u + \epsilon$, $\epsilon \sim N(0, \sigma^2)$.
It assumes a Gaussian process prior over $f$, $f \sim GP(0, k)$, which is approximated with basis functions, $f(x) = \sum_k q_k \phi_k(x)$, for instance random Fourier features.
The essence of the paper is that in the above setup, the observation model can be written as a linear model $z = \Phi q + \epsilon$, where $\Phi = [\langle v_j, \phi_k \rangle]_{i,j}$ is the inner product matrix of the basis elements $(\phi_k)$ and the solutions $(v_j)_j$ of the adjoint equation, $L^* v_j = h_j$, where $h_j$ is the observation functional of the $j$th data point.
This linear model $z = \Phi q + \epsilon$ can be used to infer $q$ from $z$ with standard Gaussian inference tools.


**Questions:**

See "weaknesses" above.

**Limitations:**

-

**Strengths And Weaknesses:**

## Strengths
* The topic is very relevant
* The methodology is sound and could be useful
* The experiments suggest that the method works
* The related work is more or less covered adequately (with some open questions, see below)


## Weaknesses
While the methodology itself is sound,
I am struggling with the premise of this work.
More specifically, I have questions about related work:

* Line 58f states: "All of these approaches require multiple forward solves of the forward problem", but the proposed algorithm also requires multiple forward solves ($n$ solves of the adjoint equation for $n$ data points). For very few data points, this might be cheaper than an MCMC approach, but competing algorithms (see next point) don't seem to require multiple simulations.

* It would be great if the paper discussed why one cannot/should not replace $\Phi q$ with $H L^{-1} f$ for some, say, finite element discretisation of $L$. This seems to be a common approach in the literature on statistical finite element methods; refer to, e.g., Eqs. (21) and (22) in [1].  It is similar to SPDE approaches for Gaussian processes. I think positioning itself against this class of ideas would improve the manuscript. Or have I missed something?

* Section 2 ("Related work") explains how an approach with Green's functions is often not feasible, because Green's functions are difficult to find and potentially unstable to apply. While I don't disagree with this statement, I would appreciate it if the experiments would include a case study in which the Green's function approach cannot be used: the ODE example seems to fit the setup from Alvarez et al.; the advection-diffusion equation appears to also admit a Green's function (see e.g., Eq. 4-3-13 in [2]; however, its representation is quite complicated)




**References**

[1] Girolami, Mark, et al. "The statistical finite element method (statFEM) for coherent synthesis of observation data and model predictions." Computer Methods in Applied Mechanics and Engineering 375 (2021): 113533.


[2] Xu, Zhanjie, Wolfgang Breitung, and J. R. Travis. Green's function method and its application to verification of diffusion models of GASFLOW Code. Karlsruhe, Germany: Forschungszentrum Karlsruhe, 2007.

---

> ### Author Response · Authors · 2022-08-02
> **Response to weaknesses 1 to 3**
>
> We’d like to begin by thanking the reviewer for their time and thoughtful comments. Below we have attempted to address their questions and concerns.
>
> 1. Why not use a method which requires fewer forward (or backward) solves?
>
> For large numbers of features, MCMC would likely still struggle to infer $f$ in the large $n$ case, as we show that for $M=50$ random Fourier features, the MCMC method did not converge after 20000 iterations, and so for the required hundreds to thousands of bases potentially required to approximate complicated 3D problems, MCMC would not be suitable.
>
> 2. Why not position this work alongside statistical finite element methods?
>
> The proposed idea of substituting $\Phi q$ with $HL^{-1}f$ appears to assume that the distribution over $f$ is known, but a key part of the adjoint method is inferring the posterior distribution of $f$ from observations. In the paper by Girolami et al. referenced by the reviewer, the focus is on inferring the system response, rather than the random forcing or diffusion terms. In particular, these random elements have known distributions and the method is focussed on determining the posterior of the system response, finite element component, model misspecification and and noise. The paper gives an example of how one might determine the the posterior distribution of the random diffusion term, and it requires 50000 iterations of an MCMC algorithm. One of the key aspects of the adjoint method is its ability to infer the posterior of the forcing function without using MCMC.
>
> The methods in the paper cited are very interesting and we thank the reviewer for bringing them to our attention. However, we believe that the adjoint method is attempting to solve a different problem.
>
> 3. Give examples of when Green's functions would not be feasible to use.
>
> It should be noted that the adjoint method extends to other linear operators beyond just differential operators, so there are scenarios where it wouldn't be possible to use the Green's function approach. In response to questions from another reviewer we have tested our approach on shift operators (a non-differential linear operator). We would be happy to include this example in the revised supplementary material. We also intend to include a comparison between the Green's function approach and the adjoint method applied to the second order ODE and hope this would abate the reviewer's concerns.

---

> > ### Comment · Reviewer_bJxr · 2022-08-03
> > **Response to response**
> >
> > Thank you for the clarifications. I will revise my review accordingly.

---

> > > ### Author Response · Authors · 2022-08-05
> > > **Thank you and a note on applications of the method**
> > >
> > > Thank you for revising your review score. We note that one of the topics you raised was the premise or purpose of this approach. To help further motivate its utility, particularly around source identification, we have added to the paper an application to a real dataset, in which we can infer *with uncertainty* the location of the pollution source with a calculation lasting just a few minutes, without needing to derive the PDE's equivalent Green's function. A summary of the experiment is below. We hope you find this addition strengthens the paper and demonstrates the power and utility of the approach. Thank you again for taking the time to review our paper.
> > >
> > > The experiment:
> > > To test the approach using data from a physical experiment, we used the Round Hill II advection/diffusion experiment [1]. In this study, researchers deployed 183 midget impingers for measuring sulphur dioxide in three partial concentric rings, 50m, 100m and 200m downwind from the release site, spanning $69^\circ$. A constant source of sulphur dioxide (releasing approximately 5-10 $g s^{-1}$) was used over a ten minute period, during which the impingers took measurements of average concentration over 30 seconds, 3 minutes and 10 minutes. The average wind speed and direction was recorded (2.14 $m s^{-1}$). We modelled this with our approach over a $250m \times 250m$ domain spanning 13 minutes. We tested two aspects of our model's capabilities. First: Source attribution. The model's mean source prediction was roughly flat except for a peak approximately 45m downwind of the true release site. This discrepancy is expected as the true dataset contained a point source while our model had a GP prior (with EQ kernel and lengthscale of 10m) over the source. This leads to an inferred broader source, slightly closer to the ring of sensors. The second test was predicting the SO${}_2$ concentration: We removed the middle (100m) ring of sensors then tried to predict their measurements. For comparison we used a Gaussian process with a length-scale of 30m (and 30s) to predict the concentration. We found it useful to threshold the concentrations to be non-negative. Our model performed considerably better than the GP model. For the three measurement periods, the results were:
> > >
> > >           MSE / (mg/m^3)^2
> > >          Our Model    GP Model
> > >     30s   20180        21749
> > >     180s   3766        13086
> > >     600s   3946         8506
> > >
> > > We will include these results, and associated figures in the final paper.
> > >
> > > [1] Cramer, H. E., and F. A. Record. "Field studies of atmospheric diffusion and the structure of turbulence." American Industrial Hygiene Association Quarterly 18.2 (1957): 126-131.

---

### Official Review · Reviewer_pYjt · 2022-07-14

**Rating:** 7
**Confidence:** 3
**Soundness:** 3 good
**Presentation:** 3 good
**Contribution:** 3 good

**Summary:**

This paper concerns statistical inference in linear systems. A typical approach uses an integral formulation of the problem via Green's function of the differential operator. The approach taken in this paper is to instead use adjoints turning the problem into a linear model and then use a reduced-rank Gaussian process for posterior approximation. In more detail, once the adjoint of the linear operator is found, n (number of observations) equations composing a linear system are set up and the unknown parameters are solved for using the standard least squares approach.


**Questions:**

In line 256, you say "compute the posterior exactly" but doesn't the adjoint method rely on a truncated basis expansion?

The adjoints you have used in your models are easy to derive analytically, are there any cases where it is trickier doing this? Are there any models where you can't use the adjoint method and have to use MCMC? I don't have a good understanding of the scope of this new method compared to the old ones.

Why are adjoint solves so fast compared to the forward problem, or at least it seems this way? The use of a single MH step requires two (or one, if the previous one is saved) likelihood evaluations. At the same time in section 4.3 you mention that you did posterior inference for 100 observations, i.e. requiring 100 adjoint solves, in the time it took you to do a single MH step. Why is your MH so inefficient? I understand that the cost of the adjoint solves, once paid, leads to a nice linear system setup and posterior approximation and this may well win over MCMC when the MCMC has to be run for a long time - which is pretty much always. However, I don't think the comparison should be this optimistic.

Finally, you say that M = 10 leads to a poor approximation, but you run your Metropolis sampler with it?


**Limitations:**

Yes.

**Strengths And Weaknesses:**

Overall, I think this is a very nice paper, and seems like a clean approach to inference in the class of problems the paper addresses. The solutions produced enjoy stability and are conceptually easier to work with and require no sophisticated MCMC methods. In this sense, I believe it is a worthwhile addition to the literature. I am generally happy with the paper, with the only note is that the MCMC comparisons are not done particularly carefully, even so, there is enough evidence in favour of the presented method at least on the demo problems.

---

> ### Author Response · Authors · 2022-08-02
> **Response to questions 1 to 4**
>
> We’d like to begin by thanking the reviewer for their time and thoughtful comments. Below we have attempted to address their questions.
>
> Q1. Why do we call the posterior "exact"?
>
> We compute the posterior exactly given the truncated basis expansion. I.e. Our method allows us to write the posterior exactly in terms of its mean and standard deviation, as opposed to estimating the posterior using MCMC. We are happy to clarify this in the revised draft of the paper.
>
> Q2. Are there cases where it is difficult to derive the adjoint? Are there cases when one couldn't use an adjoint?
>
> For many common problems the adjoint is either easy to derive or is well known. For linear algebraic problems of the form $Ax=b$, the adjoint is the transpose of the cofactor matrix $C$ of $A$ (i.e. the adjoint of $A$ is $C^\top$). For linear ODE and PDE problems, Estep (2004) provides a general formula which can be used to derive the adjoint system. In our examples the initial and boundary conditions are homogeneous, but the general formula given by Estep can account for heterogeneous ICs and BCs relatively easily.  Our method is not applicable to systems with non-linear initial and boundary conditions, or non-linear operators.
>
> Q3. Why is our Metropolis Hastings algorithm so inefficient?
>
> The Metropolis Hastings algorithm is comparatively inefficient due to the matrix multiplication required to compute the forcing function $f$ from a given $q$ using the basis vectors (see equation 8 in the paper). In the adjoint method this only needs to be calculated once, but is required at each step of the MH algorithm. Calculating the source from $q$ takes around $10\times$ longer than a single run of the forward (or adjoint) problems. This is true in both the ODE and PDE cases. We are happy to add a comment to this effect to explain the disparity between the MCMC and adjoint method speed.
>
> Q4. Why use $M=10$ if it leads to poor approximation?
>
> We have used $M=10$ in this case as a toy example to show that the MCMC and adjoint method converge to the same distribution for $q$, given a set of basis vectors. Ideally for this example we would have used more, but for $M=50$ the MCMC did not converge after $20000$ iterations.
>
> References: Estep, Donald. "A short course on duality, adjoint operators, Green’s functions, and a posteriori error analysis." Lecture Notes (2004).

---

### Meta-Review · Area_Chair_eNhf · 2022-08-23

**Recommendation:** Accept
**Confidence:** Less certain

**Metareview:**

The paper looks at a method for inference in Latent Force Models that uses the adjoint method to help out with the inference. Three of the reviewers describe the work as easy to read, sound, relevant and useful.

Some of the reviewers lament the lack of more experiments - I would have loved to see the results on the spatial experiment described in the discussion. The authors did add one experiment as part of the rebuttal - I urge them to add this to the manuscript.

One reviewer complained about the 'mathiness' of the presentation. They gave an overall score that I don't feel reflects the overall quality of the paper, and I'm inclined to discard it. Yet, I urge the authors to consider whether all of the terms used are maximising the accessibility and therefore impact of the paper. If Banach spaces are essential to the work, please explain why. If they are a small technical necessity, but unimportant to understand the main ideas, I suggest you relegate them to a formal proof in the appendix.

**Award:**

No

---

### Decision · Program_Chairs · 2022-09-14

Accept